# A quantitative genetic model of background selection in humans

**Vince Buffalo**[1,2]*, **Andrew D. Kern**[2]

**1** Department of Integrative Biology, University of California, Berkeley, Berkeley, California, United States of America, **2** Institute of Ecology and Evolution and Department of Biology, University of Oregon, Eugene, Oregon, United States of America

\* vsbuffalo@gmail.com

## Abstract

Across the human genome, there are large-scale fluctuations in genetic diversity caused by the indirect effects of selection. This "linked selection signal" reflects the impact of selection according to the physical placement of functional regions and recombination rates along chromosomes. Previous work has shown that purifying selection acting against the steady influx of new deleterious mutations at functional portions of the genome shapes patterns of genomic variation. To date, statistical efforts to estimate purifying selection parameters from linked selection models have relied on classic Background Selection theory, which is only applicable when new mutations are so deleterious that they cannot fix in the population. Here, we develop a statistical method based on a quantitative genetics view of linked selection, that models how polygenic additive fitness variance distributed along the genome increases the rate of stochastic allele frequency change. By jointly predicting the equilibrium fitness variance and substitution rate due to both strong and weakly deleterious mutations, we estimate the distribution of fitness effects (DFE) and mutation rate across three geographically distinct human samples. While our model can accommodate weaker selection, we find evidence of strong selection operating similarly across all human samples. Although our quantitative genetic model of linked selection fits better than previous models, substitution rates of the most constrained sites disagree with observed divergence levels. We find that a model incorporating selective interference better predicts observed divergence in conserved regions, but overall our results suggest uncertainty remains about the processes generating fitness variation in humans.

## Author summary

Across the human genome, there are large-scale fluctuations in genetic diversity caused by the indirect effects of selection. This "linked selection signal" reflects the impact of selection according to the physical placement of functional regions and recombination rates along chromosomes. Previous work has shown that purifying selection acting against the steady influx of new deleterious mutations at functional portions of the genome shapes patterns of genomic variation. To date, statistical efforts to estimate purifying selection

**Data Availability Statement:** All code from bgspy and our Jupyter Lab (Kluyver et al. n.d.) notebooks for analysis are available on GitHub (https://github.com/vsbuffalo/bprime). The main model fits are available as Python Pickle objects on Data Dryad

repository (https://doi.org/10.5061/dryad.qnk98sfnv).

**Funding:** This research was supported by National Institute of Health awards R35GM148253 and R01HG010774 to ADK. The funders had no role in study design, data collection and analysis, decision to publish, or preparation of the manuscript.

**Competing interests:** The authors have declared that no competing interests exist.

parameters from linked selection models have relied on classic Background Selection theory, which is only applicable when new mutations are so deleterious that they cannot fix in the population. Here, we develop a statistical method based on a quantitative genetics view of linked selection, that models how polygenic additive fitness variance distributed along the genome increases the rate of stochastic allele frequency change. By jointly predicting the equilibrium fitness variance and substitution rate due to both strong and weakly deleterious mutations, we estimate the distribution of fitness effects (DFE) and mutation rate across three geographically distinct human samples. While our model can accommodate weaker selection, we find evidence of strong selection operating similarly across all human samples. Although our quantitative genetic model of linked selection fits better than previous models, substitution rates of the most constrained sites disagree with observed divergence levels. We find that a model incorporating selective interference better predicts observed divergence in conserved regions, but overall our results suggest uncertainty remains about the processes generating fitness variation in humans.

## Introduction

The continual influx of new mutations into populations is the ultimate source of all adaptations, but the vast majority of mutations either do not affect fitness or are deleterious. Natural selection works to eliminate these deleterious mutations from the population, thus we expect them to appear at low frequencies within populations [1], and be less likely to fix between lineages. Conserved genomic regions reflect the product of hundreds of millions of years of evolutionary optimization; thus the overwhelming majority of segregating variation in these regions will have deleterious fitness effects. Consequently, a good predictor of whether a new mutation will reduce fitness is if it occurs in a region of the genome that has been conserved over phylogenetic timescales [2, 3]. Moreover, segregating rare variation in these regions is responsible for a significant proportion of the genetic contribution to phenotypic variation and disease in humans [4–7].

Selection on both beneficial and deleterious variants perturbs the allele frequencies of neighboring linked sites, a phenomenon known as linked selection [8–12]. Since deleterious variation is clustered in functional portions of the genome, we expect linked selection to reduce levels of diversity around evolutionarily constrained segments (e.g. coding sequences, splice sites, regulatory elements, etc.). The genomic arrangement of these conserved regions coupled with heterogeneous recombination rates create a large-scale spatial signal of linked selection of genetic diversity along chromosomes. Since genome-wide recombination maps and functional annotations are available for many species, there has been consistent effort to fit models of linked selection to patterns of diversity. This general approach provides estimates of population genetic parameters such as the strength of selection and the deleterious mutation rate [13, 14], and potentially distinguishes the roles of positive and negative selection and estimate the rate of beneficial mutations [15, 16]. In humans, previous work has shown that negative selection plays the dominant role in shaping megabase-scale patterns of diversity, with positive selection having a nearly negligible impact [16].

Prior work to model the reduction in linked diversity due to deleterious mutations has largely relied on the classic Background Selection (BGS) model [8, 11, 13, 17]. While the BGS model has been successful in fitting many patterns of diversity, some of its simplifying assumptions may distort inferences about the selective process. First, since fixation probabilities ultimately depend on the product of the deleterious selection coefficient ($s$) and population size

($N$), the efficacy of selection depends on past population sizes. Unfortunately, accommodating such demography into analytic models of purifying selection remains an open, difficult problem [18, 19] though simulation-based inference may be a route forward [20]. Second as the BGS model builds off classic models of mutation-selection balance [21, 22], it assumes that new mutations are sufficiently deleterious that they are invariably driven to loss. Under this assumption, the effect of selection is well-approximated by simply rescaling the neutral coalescent by a reduction factor known as $B = {N_e}/{N}$ [23]. However, this simple rescaling approach is not appropriate across parts of parameter space that are relevant to natural populations [24, 25]. In particular, the BGS model cannot accommodate the possibility of weakly deleterious mutations (those with fitness effects $2Ns \leq 1$) reaching fixation, which leads to incorrect predictions of diversity levels as the strength of selection diminishes. Finally, the classic BGS model assumes that the selective dynamics at one site are not impacted by selection at other positions, i.e. no selective or "Hill–Robertson" interference [24, 26, 27].

In this work, we use another class of linked selection models that derive from quantitative genetics to address limitations of the classic BGS model [28–31]. These models consider how polygenic fitness variance spread along the genome increases the variance of stochastic allele frequency change, as alleles become randomly linked to fitness backgrounds over time and their frequency trajectories are perturbed by selection at other sites. While these models can theoretically accommodate additive fitness variance from any source as long as its rate of change is not too rapid, we focus specifically on a deleterious-mutations-only model of fitness variance from [29]. This model is identical to BGS when selection against deleterious mutations is strong, but it also correctly predicts the reduction in diversity when selection is weak by jointly predicting the deleterious substitution rate. We extend the Santiago and Caballero (hereafter the SC16) model of the negative selection process so that it can be fit using a composite likelihood approach to patterns of genome-wide diversity, according to the spatial distribution of genomic features that could harbor deleterious fitness variation. Using forward simulations, we show this model leads to more accurate estimates of the distribution of fitness effects (DFE) under weak selection. We apply our composite-likelihood method to human population genomic data and provide new parameter estimates of the genome-wide impact of purifying selection in humans. We show that our new method is better able to predict the patterns of diversity along human chromosomes than previous models. However, our model leads to predictions of the deleterious substitution rate that disagree with observed levels of divergence. We discuss the potential causes and implications of such discrepancies and what it might mean for future efforts to fit linked selection models to genomic patterns of variation.

## Theory

Our work extends quantitative genetic models of linked selection ([28–31]; see also the Appendix of [8]), which approximate the reduction in genetic diversity due to linked selection in terms of polygenic additive fitness variance ($V_A$). These models are approximations to fairly complicated selection dynamics; in reality, background selection cannot be fully summarized by simply rescaling effective population size across all of parameter space [25, 32, 33]. Consequently, we use extensive, realistic forward simulations (in the next section) to demonstrate the validity of these models in the region of parameter space that the human genome occupies.

Here we review the relevant theory before introducing our genome-wide extension. These linked selection models stem from Robertson [31], which in essence describes how polygenic additive fitness variation increases the pace of stochastic allele frequency change, thus reducing effective population size. At the individual level Robertson considered, selection generates an autocorrelation in fecundity as offspring from large families tend to beget many descendants

themselves (and likewise with small families) when fitness is heritable. This same across-generation autocorrelation occurs at the genomic level due to linkage [28, 34], as the perturbations to a neutral allele's trajectory from its particular fitness background tend to occur in the same direction across generations until the background recombines off. Quantitative genetic models such as Santiago and Caballero's [28] quantify the total impact of the autocorrelation generated by selection in terms of what we think of as a *fitness-effective* population size $N_f$ (to differentiate it from the *drift-effective* population size, which is the size of the ideal population when there is no fitness variation).

The key insight is that in the long run, the steady presence of additive genetic fitness variance ($V_A > 0$) contributes an extra source of variance in offspring number beyond the variance expected under pure drift [35]. However, because heritable fitness variation generates across-generation autocorrelation, the cumulative effect of this fitness variance on the variance in allele frequency change is inflated by a factor of $Q^2$. Intuitively, the product $V_A Q^2$ represents the expected total variance in reproductive success a neutral mutation experiences over its lifetime in a system with weak selection at linked sites.

Following Robertson and Santiago and Caballero [30, 31], we define the fitness-effective population size $N_f$ by including the total additional variance created by heritable fitness into Wright's equation [35] for effective population size,

$$N_f = \frac{N}{Q^2 V_A + 1} \tag{1}$$

(c.f. [30, 31]; see S1 Text Section 1 for a proof). The benefit of modeling linked selection with Robertson's forward-time model is that the inflation factor under weak selection is invariant with respect to the particular fitness background (i.e. high or low fitness backgrounds are exchangeable in this model) the neutral allele becomes stochastically associated with. By contrast, modeling diversity levels under linked selection backwards in time requires tracking the particular associated fitness backgrounds, as coalescence rates experienced by a lineage are not invariant with respect to their fitness background, i.e. high and low fitness backgrounds are not exchangeable.

Eq (1) is general, since different modes of selection and linkage can be accommodated by different expressions for $V_A$ and the inflation factor $Q^2$ [28, 30]. When fitness variation has a multiplicative polygenic basis, as is often assumed for genome-wide selection processes, the fitness-effective population size experienced by an arbitrary neutral site under the influence of all $S$ linked regions is,

$$N_f \approx N \exp\left(-\sum_{i=1}^{S} V_{A,i} \frac{Q_i^2}{2}\right) \tag{2}$$

where the factor of one-half comes from ignoring the short-lived associations with the homologous chromosome, which have a weak effect on the focal allele (see S1 Text Section 1.3). In our genome-wide model, we consider the summation in Eq (2) over non-overlapping contiguous, putatively conserved segments $i \in \{1, 2, \ldots, S\}$ (e.g. exons, splice sites, etc.) each undergoing selection such that segment $i$ contributes additive fitness variance $V_{A,i}$ to the total additive genetic fitness variance. The impact this segment's fitness variance $V_{A,i}$ has on the fitness-effective population size is mediated by the autocorrelation term $Q_i$, which is a decaying function of the recombination rate between the segment and focal neutral allele. Specifically, the autocorrelation function for a neutral allele associated with segment $i$ is $C(t) = [(1 - r_i)(1 - \kappa_i)]^t$, where $r_i$ is the recombination fraction to the segment and $\kappa_i$ is the rate that the associated fitness variance decays due to selective dynamics. Then, the cumulative autocorrelation over the lifespan

of the allele is,

$$
\begin{aligned}
Q_i \quad &= 1 + \sum_{t=1}^{\infty} [(1 - r_i)(1 - \kappa_i)]^t \\
&= \frac{1}{\kappa_i + r_i(1 - \kappa_i)}.
\end{aligned}
\tag{3}
$$

(see S1 Text Equation 20). This general equation can accommodate models of polygenic selection as long as the equilibrium additive fitness variation $V_{A,i}$ can be specified and the change in variance due to selection can be approximated as a geometric decay, i.e. $\Delta V_{A,i} = -\kappa V_{A,i}$ [28, 36–38]. This is usually a reasonable assumption since within-generation selection removes a fraction of phenotypic variation from the population, and some fraction of that is additive genetic variation [36, 37, 39].

The remaining required expressions are for the equilibrium additive fitness variance $V_A$ and the decay rate in associated fitness $1 - \kappa$. Fitness variance could arise from beneficial or deleterious alleles, but given prior work has found selection against new deleterious mutations in conserved regions plays a dominant role in shaping genome-wide patterns of diversity and divergence [14, 16], we focus specifically on purifying selection. We imagine a mutation-selection process that creates fitness variation as deleterious mutations enter a population at rate $\mu$ per basepair per generation in a conserved region of $L$ basepairs, such that the region-wide per generation diploid mutation rate is $U = 2\mu L$. Each mutation imposes a selective cost of $s$ in heterozygotes and $2s$ in homozygotes, and fitness effects are multiplicative across sites.

Under this selection model, the additive genic fitness variance created by a new mutation (at frequency $x = 1/2N$) is $2s^2 x(1 - x) \approx s^2/N$. For the entire population of $2N$ chromosomes, the mutational variance input each generation in segment $i$ is $V_{M,i} \approx U_i s^2$ where $U_i = 2\mu L_i$ is the diploid mutation rate per generation within the segment. Under the mutation-selection balance assumed by classic BGS theory, an $L_i$-basepair segment has equilibrium additive genetic variance $V_{A,i}^{BGS} \approx U_i s$ (see S1 Text Equation 40) and thus $\kappa_i^{BGS} = s$. Substituting $V_{A,i}^{BGS}$ and $\kappa_i^{BGS}$ in Eq (2) and simplifying, we have

$$
N_f = N \exp \left( -\sum_i^s \frac{\mu L_i}{s(1 + r_i(1 - s)/s)^2} \right)
\tag{4}
$$

which is identical to the genome-wide model of background selection used in previous studies [14–16]. Thus, the classic background selection model is a special case of the more general theory of Santiago and Caballero [29], which they had shown previously [28].

However, when new mutations are only weakly deleterious, they can drift up in frequency before their eventual loss or fixation. At this point, the number of deleterious mutations per haplotype is no longer well-approximated by the deterministic mutation-selection-balance theory, and their dynamics are strongly influenced by stochastic perturbations due to both drift and linked selection. In this weak selection regime, classic BGS theory no longer accurately predicts levels of linked diversity [11, 24, 25, 40]. Moreover, selection against weakly deleterious mutations alters the topology of genealogies, such that they are no longer well-approximated by a rescaled neutral coalescent as assumed under the classic BGS model [41–43]. To further complicate matters, the distribution of the number of deleterious mutations (and its corresponding fitness distribution) is no longer a stationary Poisson distribution, instead becoming a traveling wave [33, 44, 45] towards increased numbers of deleterious alleles per chromosome and reduced mean population fitness. In asexual populations, each click of "Muller's ratchet", which is the stochastic loss of the least-loaded class [46, 47], on average

leads to one deleterious substitution [43]. Unfortunately, determining the rate of deleterious substitutions is another difficult problem [40, 43, 45, 48, 49] related to Hill–Robertson interference [27, 50].

Quantitative genetic models of linked selection approximate an equilibrium $V_A$ under both strong and weakly deleterious mutations by concurrently modeling the rate of fixation of deleterious alleles in the region, $R$ (for clarity, we consider just a single segment and omit the index $i$). Santiago and Caballero [29] suggest that equilibrium fitness variance is lower than predicted by $V_A^{BGS}$ once weakly deleterious mutations begin to have an appreciable rate of fixation $R > 0$ per generation in the region. The substitution rate $R$ decreases fitness variance since each substitution removes a segregating site and thus its contribution to fitness variance. Thus the steady-state additive genetic variance of fitness under mutation and negative selection is,

$$V_A = (U - 2R)s. \tag{5}$$

where the condition $V_A \geq 0$ is met when the probability of fixation is less than or equal to the neutral fixation probability of $^1/_{2N}$, as is true for all deleterious mutations. This equation describes the equilibrium additive genetic fitness variance as the balance of the flux of new variation in to the population from deleterious mutations, and the removal of variation due to their substitution (and the decline in mean population fitness). When $R = 0$, selection is so strong deleterious alleles cannot fix, and the equilibrium fitness variation is due entirely to young rare mutations before their extinction $V_A = V_A^{BGS} \approx Us$. Santiago and Caballero derive Eq (5) through Fisher's Fundamental Theorem of Natural Selection, but we find an alternative proof ([43]; see S1 Text Section 1.9). We also find that the steady-state additive genic variance in Eq (5) results from diffusion models with a flux of mutations into discrete sites ([51]).

While using Eq (5) in Eq (1) leads to a prediction for the fitness-effective population size $N_f$, closed-form expressions for the deleterious substitution rate $R$ have generally been hard to find [29, 43, 45, 48]. A key insight of Santiago and Caballero [29] is that the deleterious substitution rate with linked selection can be approximate by using the probability of fixation $p_F(N_f, s)$ [52, 53] using the rescaled fitness-effective population size, i.e. $R = NUp_F(N_f, s)$. Given this equation for the substitution rate and Eq (2) for $N_f$ under linked selection, we have a system of two non-linear equations that can be solved numerically for $N_f$ and $R$ for each segment (again, omitting the segment index $i$ for clarity),

$$N_f = N \exp\left(-(U - R)s\frac{Q^2}{2}\right) \qquad \text{fitness-effective population size equation} \tag{6}$$

$$R = \frac{4N_f Us}{\exp(4N_f s) - 1} \qquad \text{substitution rate equation} \tag{7}$$

We denote the solutions to these equations, which represent equilibria under mutation-selection-drift process, as $\widetilde{N}_f$ and $\widetilde{R}$. These equilibria also imply an equilibrium level of additive fitness variation $\widetilde{V}_A$ in the segment, which are used to calculate the reduction factor $B(x) = {}^{N_f}/_N$ at any other genomic position $x$ (see Methods Section Calculating the reduction maps). In our inference method, described in Methods Section Composite likelihood and optimization), we extend these equations to handle a distribution of selection coefficients, and multiple feature classes. During inference, we also consider an alternate "local rescaling" model that sets the $N$ in the fitness-effective population size equation to $B(x)N$ and re-solves these equations. This alternative model approximates the impact of other segments on each focal segment's selection dynamics by using the local effective population size implied by the estimated B map.

We note that solutions to these equations do not accurately model the substitution rate under all regions of parameter space [25, 49, 54]; in the next section we show through extensive simulations that this approximation works quite well under human parameters.

## Results

We provide two main classes of results. First, we show simulation results which demonstrate the accuracy of the SC16 model over the BGS approximation across the parameter space, as well as validations of the composite likelihood strategy we use to fit the SC16 model. Second, we provide fits of our method to human genome data, where we show comparison of models fit using different annotations, the estimated DFEs, and predictions of the deleterious substitution rate.

### Simulation validation of theory and methods

Given that modeling the interplay of mutation, drift, and linked selection under both weak and strong selection has proven to be a difficult problem, we first sought to verify the SC16 theory and our genome-wide extension with three levels of simulations: forward simulations of purifying selection in a region, chromosome-scale forward simulations of purifying selection, and simulations of a "synthetic genome" (i.e. by combining independently simulated chromosomes) to test our composite-likelihood method based on this theory.

**Simulations of a segment under purifying selection.** Our first set of forward simulations was to ensure that the SC16 model adequately captures selective dynamics in a single 100 Mbp basepair region under selection, across a variety of mutation rates and selection coefficients (see Methods Forward simulations). We find a close correspondence between the observed and predicted reductions in effective population size $B = {N_f}/{N}$ over all selection and mutation parameters including weak selection (Fig 1A), in contrast to classic BGS theory. Furthermore, to investigate whether this accuracy was caused by the model correctly predicting the equilibrium fitness variance and substitution rate, we also measured these throughout the simulation. Again, we find diploid SC16 theory accurately predicts both the deleterious substitution rate (Fig 1B) and the genic fitness variance (Fig 1C).

Moreover, these simulations provide intuition about the underlying selection process. When mutations are strongly deleterious, there is no chance they can fix, and the substitution rate is zero (Fig 1B for $2Ns > 1$). In this strong selection regime, the additive genic fitness variation closely matches the theoretic deterministic equilibrium of $V_A = Us$ (dashed gray line, Fig 1C) However, around $2N_e s \approx 1$, the substitutions begin to occur as $p_F$ moves away from zero. When this occurs, each fixation eliminates variation, and the equilibrium variation diverges from the deterministic mutation-selection equilibrium (Fig 1C).

**Chromosome-wide simulations and models of negative selection.** Given the accuracy of the SC16 model in predicting the reduction factor $B$ and the deleterious substitution rate for a single segment under general mutation-selection processes, we next extended their model so that it could be fit to patterns of windowed genome-wide diversity through a composite likelihood approach. Our software method `bgspy` numerically solves Eq (6) to compute the equilibrium additive genic fitness variance ($\widetilde{V}_a$) and the deleterious substitution rate ($\widetilde{R}$) across grids of mutation rates and selection coefficients. This is done for each pre-specified segment in the genome (potentially tens of millions of small regions, which depend on the particular annotation of putatively conserved regions used) that may be under purifying selection (e.g. coding sequences or UTRs). We call the set of theoretic predicted reductions across these grids the B' maps (to distinguish them from McVicker's B maps [14]; these can be used to find the equilibrium reduction factor $B(x)$ for any genomic position $x$.

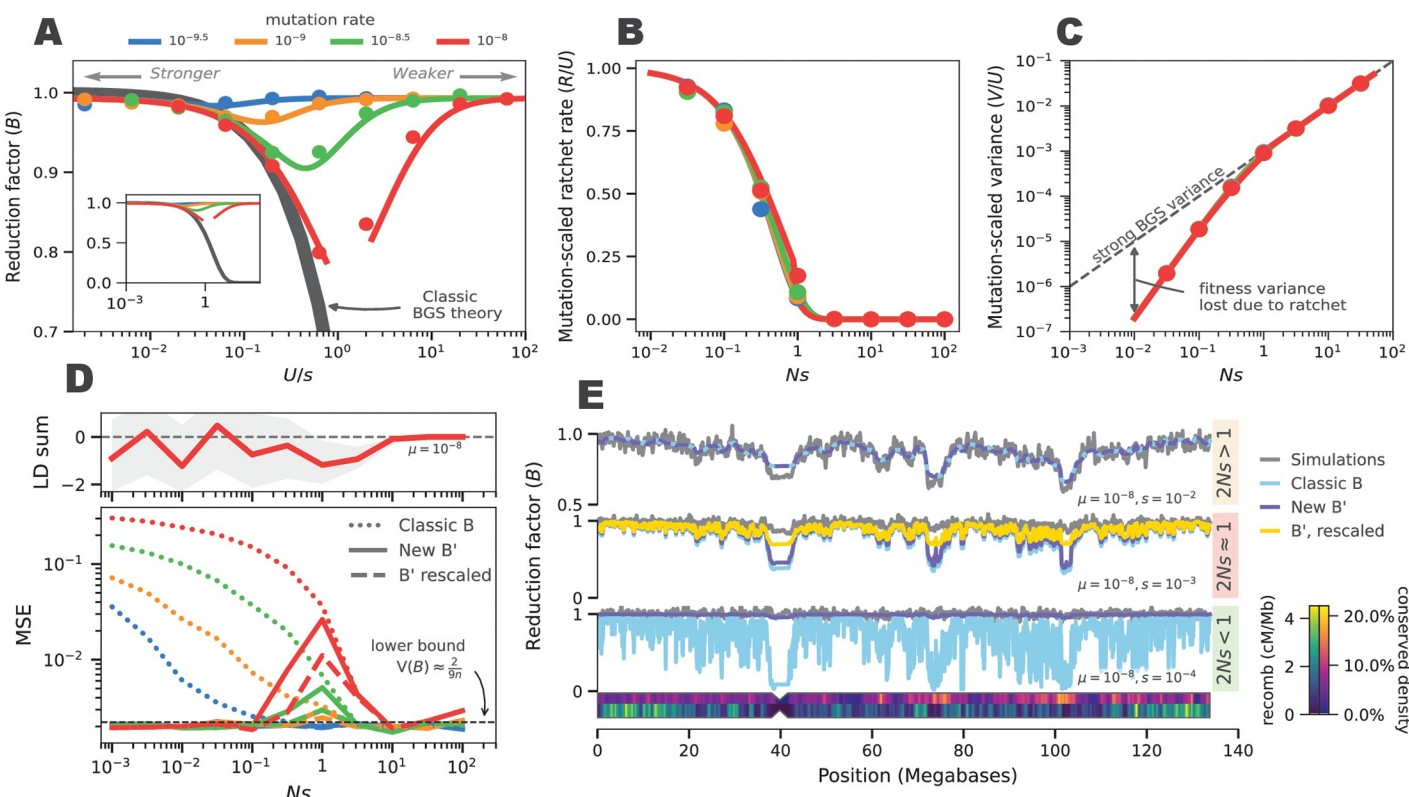

**Fig 1. Santiago and Caballero (2016) theory models the weak selection regime better than classic BGS theory.** (A) The predicted reduction factor under classic B theory (dark gray line) and the diploid SC16 model (colored lines corresponding to mutation rate) compared to average reduction across 10,000 simulation replicates (points). The inset figure is zoomed out to show extent of disagreement under classic BGS. (B) The predicted deleterious substitution rate under the SC16 model, scaled by mutation rate (colored lines) compared to the substitution rate estimated from simulation (points). When $2Ns > 1$, the substitution rate is near zero. (C) The genic variance from simulations (points) against the predicted variance under the SC16 model (colored lines). As substitutions begin to occur, the genic variance is decreased from the level expected under strong BGS (dashed line). (D, bottom) The mean squared error (MSE) between whole-chromosome simulations and predicted classic B (dots), new B' (solid), and locally rescaled B' (dashed) for different mutation rates (colors). Locally rescaled B' (yellow lines) are omitted for clarity in the top and bottom rows, since they are identical to B'; Local rescaling only impacts B' in the $2Ns \approx 1$ domain. The dashed horizontal line is the approximate theoretic minimum MSE. (D, top) The build-up of negative linkage disequilibria around $2Ns = 1$ in whole-chromosome simulations shown in the bottom panel. (E) The average B map from 100 chromosome 10 simulation replicates (gray) against different predictions, for parameters that correspond to $2Ns < 1$, $2Ns = 1$, and $2Ns > 1$. The chromosome shows the density of conserved sites and recombination map used in simulations.

We validated our predicted B' reduction maps with realistic chromosome-scale forward simulations of purifying selection using putatively conserved regions and recombination maps for the human genome. We find that our B' maps and the classic BGS theory B maps closely match simulations when selection is strong (top row of Fig 1E), apart from slight discrepancies in low recombination regions (Fig 1E). Second, we find our theory is vastly more accurate than the classic BGS when selection is very weak ($2N_e s \ll 1$; bottom row of Fig 1E). In essence, these findings represent the facts that classic BGS theory (which as shown in (4) is a special case of the SC16 model) is accurate when selection is relatively strong and Eq (6) are accurate as $s \to 0$. Across all mutation and selection parameters simulated, the relative error of the classic B maps is 14.6% whereas the relative error in the new B' maps is 5%. Nearly all of this error is in the nearly neutral domain ($2N_e s \approx 1$ domain); for strong and weak selection, the mean squared error between simulations and B' maps is close to the theoretic lower bound of the mean squared error, $\approx {}^2/_{9n}$ set by the coalescence variance for a 10 kb region [55].

We hypothesized this error in the nearly neutral domain may be due to selective interference between segments that is not taken into account when we numerically solve Eq (6)

independently for each segment. In particular, when we numerically solve these equations, we use a fixed drift-effective population size, $N = 1,000$, corresponding to the number of diploids in the simulations. However, in reality, selection throughout the genome would lead a segment at position $x$ to experience a locally reduced effective population size of approximately $B(x)N$, which is a consequence of selective interference [26, 27, 50]. To test this, we implemented a "locally rescaled" version of the B' maps, which uses $B(x)N$ as the population size when numerically solving these equations. We use this approach because (1) iteratively solving Eq (6) for the entire genome in an inference framework is computationally infeasible, and (2) comparing the initial fits and local rescaling fits allows us to observe how incorporating the local fitness-effective population size impacts parameter estimates, if at all. We find the locally rescaled B' maps reduce the relative error from 5% to 0.4% and mean squared error (Fig 1D, dashed colored lines), but does not entirely eliminate the error in the $2Ns \approx 1$ domain (where the linkage disequilibrium build up is the highest, Fig 1D, top row).

**Validation of composite-likelihood method using forward Simulations.** Our composite-likelihood method estimates the distribution of selection coefficients for each feature type, the mutation rate, and the diversity in the absence of linked selection ($\pi_0$) by fitting the theoretic reduction map to windowed genome-wide diversity (see Methods Composite likelihood and optimization). We validated that our method can accurately estimate the selective parameters by simulation a "synthetic genome" of the first five human chromosomes (see Methods Forward simulations). We note three findings from these simulations.

First, both our implementation of classic BGS theory and our B' method accurately infer the average selection coefficient under strong selection (Fig 2, middle row). However, when selection was weak, the classic BGS model erroneously estimated strong selection and a very low mutation rate. By contrast, our B' method estimated selection coefficients much more accurately. A minor discrepancy occurs around $2Ns = 1$, likely due to the sensitivity of mutations in this region to selective interference (these results do not use local rescaling). To ease computational costs, we only simulated fixed selection coefficients and five chromosomes, and we only assessed the accuracy of average selection coefficients rather than the full estimated DFE.

Second, we find slight biases in mutation rate estimates from both our B' and the classic BGS methods (Fig 2, bottom row). However, mutation rate estimates based on our B' method are more accurate than classic BGS theory across a range of selection coefficients. Overall, this bias in estimated mutation rates suggests that benchmarking genome-wide negative selection models based on their agreement with pedigree-based rate estimates may not be appropriate. When BGS is not occurring, either due to weak selection or a low rate of deleterious mutations (Fig 2, right column), all estimates deteriorate. This is understandable, as the overall signal from linked selection weakens relative to drift-based noise. We should note, though, that this is an unlikely region of parameter space and this issue can be readily diagnosed from the low $R^2$ values. Finally, we find in additional tests that our estimates are robust to demographic expansions but inaccurate when mutations have recessive effects, since our model assumes additive effects (see S1 Text Section 5.3). We did not test the influence of population bottlenecks, since parameter estimates of out-of-Africa bottlenecked populations (CHB and CEU) did not differ much from YRI estimates (see below).

Third, we find the coefficient of determination, $R^2$, between predicted and simulated megabase-scale diversity serves as a measure of the strength of the linked selection signal in genome-wide data. $R^2$ increases with the intensity of selection against new deleterious mutations and mutation rate (Fig 2, top row). Under just drift or weak purifying selection, the variance in diversity is driven by unstructured coalescence noise along the genome and the predicted reduction map, $B(x)$, and does not fit the data well. Under very strong selection

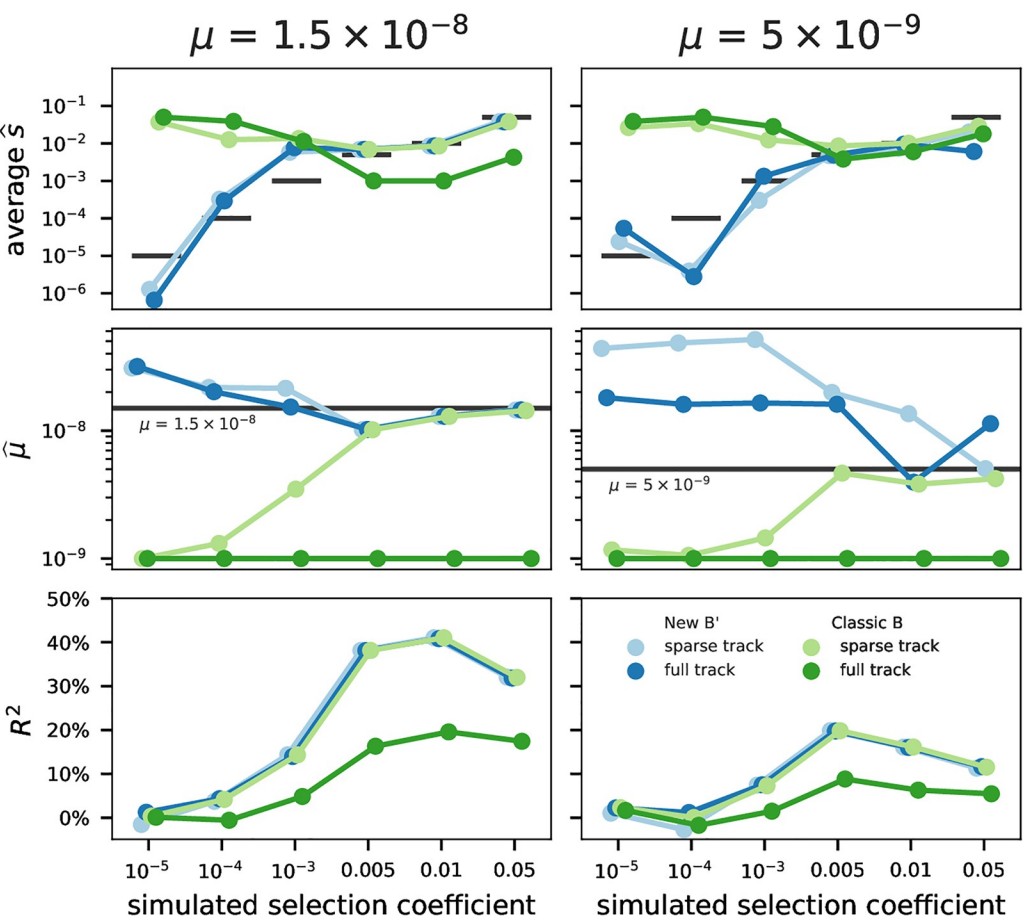

**Fig 2. Comparison of parameter estimates using classic BGS theory (green lines) with our new B' method (blue lines) across both full and sparse track types (dark versus light hue), and different mutation rates (columns).** Both classic BGS and B' methods correctly estimate strong selection coefficients when annotation tracks are sparse, but only B' can accurately estimate selection coefficients when selection is weak or full annotation tracks are used (first row). Mutation rate estimates (second row) are more accurately estimated by the B' method than classic BGS across selection parameters, but overall show slight biases. Additionally, $R^2$ between predictions and observations increases with selection intensity (third row). Overall, classic BGS methods break down as expected when full-coverage tracks are used, since it cannot accommodate weak selection and neutrality in putatively conserved regions. See Methods for details on sparse versus full tracks.

($s = 0.05$), $R^2$ is reduced; this is likely due to very strong selection having less localized effects and impacting overall genome-wide diversity [30, 31].

### Application to human genomic data

**Annotation model comparison.** Our composite likelihood method takes tracks of annotated features (an "annotation model") that are *a priori* expected to have a similar distribution of fitness effects, and estimates the overall mutation rate and distribution of fitness effects for each feature type. These annotation models specify the putatively conserved "segments" used in Eqs (2) and (4). We consider two classes of annotation models: (1) CADD-based models, which consider the top *x*% most pathogenic basepairs according to the CADD score, and (2) and more interpretable, feature-based models that includes protein coding regions, introns and UTRs, and PhastCons regions. We include PhastCons regions because they include

highly-conserved, non-coding regions known to harbor important functions [2, 56–58], that would be missed by gene feature only annotation. These two classes of annotation models have a trade-off between fine-scaled specificity to which basepairs are likely to be under negative selection, and interpretability of the DFE estimates for each feature. Finally, for each annotation model, we fit a "sparse" track version (conserved regions only) and a "full track" version (which includes another feature class called "other" that includes the rest of the genome).

Our method estimates a distribution of fitness effects (DFE) for each feature class. While CADD-based models only have a single conserved feature class (e.g. CADD 6%), feature-based models can have multiple feature classes under varying levels of selective constrain. However, overlapping features (e.g. a basepair that is annotated as both PhastCons and coding sequence) must be assigned to one category or the other. Since this assignment impacts DFE estimates, we fit both of the two alternative models. First, a *PhastCons Priority* model, where genic features that overlap PhastCons regions are classified as PhastCons, and all remaining coding basepairs are labeled as CDS. Second, a *Feature Priority* model, where all coding basepairs are assigned to CDS, and the PhastCons class catches the remaining highly-conserved non-genic regions.

In total, we fit four annotation models (CADD 6%, CADD 8%, PhastCons Priority, and Feature Priority) to high-coverage 1000 Genome data for three reference samples: Yoruba (YRI), Han Chinese (CHB), and European (CEU). We assess and compare our models according to how well they predict patterns of diversity on whole chromosomes left-out during the model fitting process (e.g. leave-one-chromosome-out, LOCO). We use the metric $R^2_{\text{LOCO}}$, which is the proportion of the observed variance in genomic diversity at the megabase scale predicted by our model on held out data. We experimented with a few smaller spatial scales (e.g. 100 kbp), but our results were consistent with previous results suggesting the human linked selection signal due to purifying selection fits best at the megabase scale [16]. Intuitively, the poorer model fits at smaller spatial scales can be understood as a result of fewer mutations and marginal coalescent genealogies being averaged over, increasing these sources of noise relative to the linked selection signal.

Overall, we find the PhastCons Priority and CADD 6% models fit equally well (Fig 2A), consistent with recent work using classic BGS theory [16]. However, we find that our models predict out-of-sample diversity levels slightly better than previous methods. For these two models, we find that our B' method predicts $R^2_{\text{LOCO}} = 67.3\%$ and $R^2_{\text{LOCO}} = 66.7\%$ of the out-of-sample variance in Yoruba pairwise diversity at the megabase-scale, respectively. By contrast, the best-fitting CADD 6% model from Murphy et al. [16] explained 60% of diversity in left-out 2 Mbp windows across YRI samples. We note that this difference could be explained by other differences in data processing, optimization, etc. For lineages impacted by the out-of-Africa bottleneck, the goodness-of-fit was lower across all models (e.g. 61.0% and 58.8% for CEU and CHB respectively in the PhastCons Priority model).

Since our method is built upon theory that fixes the weak selection problem of classic BGS theory, it should in principle fit equally well when an annotation model includes regions that are under no or little selective constraint and thus (nearly) neutrally evolving. Consequently, our B' method should fit equally well when applied to "sparse" and "full" track models, since our method in principle can accommodate weak selection and neutrality. Indeed, we find that both in-sample $R^2$ and out-of-sample $R^2_{\text{LOCO}}$ values are nearly identical across full and sparse-track models (Fig 2A and 2B, round points), which demonstrates that our method is able to deal with weak selection and that there is little more predictive power to gain from including sites considered "other" as annotations.

By contrast, full annotation models fit poorly under classic BGS theory, and lead to unreasonable parameter estimates. Additionally, when sparse annotation models contain genomic features that are likely under weak constraint (such as introns and UTRs), models fit worse under classic BGS theory than our B' method (Fig 2A). However, among the CADD annotation models, the goodness-of-fit is nearly identical between B' and classic BGS methods. This behavior is what we would expect given that the CADD models contain only the most pathogenic sites, which are *a priori* very likely under the strong selection domain under which B' and classic BGS theory agree. Finally, we note that the predicted classic B and B' maps are nearly identical under the CADD 6% model ($R^2$ = 99.99%, see S1 Text Section 5.6 for a comparison). This reflects the fact that the top 6% most pathogenic CADD sites are under strong selection, and both models are identical in this domain.

Overall, our $R^2_{\mathrm{LOCO}}$ estimates suggest our model explains up to 67% of out-sample variance in diversity of the megabase scale, even though our method assumes constant demography and homogeneous mutation rates along the genome. A worthwhile question is: how much variation *could* we expect to fit at this scale? Given that selection alters genealogies in ways beyond just decreasing mean pairwise coalescence time and populations have non-constant demography, an exact analytic answer is intractable. However, we can get an approximate idea if we assume that the residual variance $\sum_b (\pi_b - \widehat{\pi}_b)^2$ is determined entirely by the expected neutral coalescence noise around the expected coalescence time $2B(b)N$. This can be be found analytically, plugging-in our predictions for $B(b)$. This allows us to calculate $R^2_{\mathrm{coal}}$ to ballpark the theoretic variance that is capable of being explained, assuming this coalescent noise process alone (see S1 Text Section 3.6). We note that selection is expected to *decrease* the variance in coalescence times beyond a rescaled effective population size implies, thus our $R^2_{\mathrm{coal}}$ would be an underestimate under models with selection.

We find that our out-sample $R^2_{\mathrm{LOCO}}$ for the Yoruba samples ($R^2_{\mathrm{LOCO}} \approx 67\%$) is slightly above the theoretic $R^2_{\mathrm{coal}} \approx 66.6\%$. This suggests our model is in the vicinity of fitting all the signal possible, under the coalescence-only noise assumption. By contrast, for bottlenecked out-of-Africa samples, we find a larger discrepancy between $R^2_{\mathrm{coal}}$ and observed out-sample $R^2_{\mathrm{LOCO}}$. The theoretic $R^2_{\mathrm{coal}} \approx 63\% - 64\%$ for both samples, compared to the observed $R^2_{\mathrm{LOCO}} \approx 61\%$ for CEU and $R^2_{\mathrm{LOCO}} \approx 59\%$ for CHB under the PhastCons Priority model. Given that bottlenecks would act to increase the residual variance in coalescence times beyond the level implied by the effective population size, this gap would likely shrink under more realistic models or simulation-based approximations for $R^2_{\mathrm{coal}}$. Overall, this suggests that purifying selection models fit the vast majority coalescence time variation at the megabase-scale that is capable of being explained (i.e. that is not coalescent noise).

**Estimated distribution of fitness effects.** Our composite likelihood method has three sets of parameters: the expected diversity in the absence of linked selection diversity $\pi_0$, the mutation rate $\mu$, and the matrix of distribution of fitness effects **W** across the selection grid for each of the $K$ feature types. Given that the relationship between $\pi$ and the strength selection is U-shaped (i.e., see Fig 1A), we wondered whether our new B' model accommodating weak selection would fit the linked selection signal under a different combination of weak and strong selection parameters than observed previously. However, across all of our annotation models, new deleterious mutations in conserved feature classes (e.g. CADD tracks and PhastCons regions) were consistently estimated to have strongly deleterious effects (Fig 3A), consistent with previous work [14, 16]. The DFE estimates for CADD and PhastCons regions consistently places $\geq 75\%$ of mass on the largest selection coefficient we used, $s = 10^{-2}$. The CADD 6% DFE estimates imply an average selection coefficient of $\bar{s} = 0.0065$ for CEU, $\bar{s} = 0.0057$ for CHB, and $\bar{s} = 0.0079$ for YRI. Similarly, the PhastCons Priority model implies

average selection coefficient estimates of $\bar{s} = 0.0063$, $\bar{s} = 0.0059$, and $\bar{s} = 0.0077$ for CEU, CHB, and YRI respectively for PhastCons regions. Our DFE estimates for CDS under the Feature Priority model are weaker than those for non-coding PhastCons regions; this reflects the fact that around 30% of mutations to coding sequences result in a synonymous change [59] and are thus likely effectively neutral. Our results are qualitatively consistent with the U-shaped DFEs found for amino acids through Poisson Random Field method [60], but differ from other estimates based on the depletion of rare variants in functional regions [61]. Given the large differences in sample size between the present study and that of [61] as well as the differences in methodology, it is perhaps unsurprising that our results are in closer alignment with SFS approaches. However, we note that our BGS parameterization of Eq (2), excludes a role for weak positive selection; this form of model misspecification may bias our DFE estimates and our results should be interpreted in light of this.

Following previous work, our method used a grid of selection coefficients up to $s = 10^{-2}$. However, we also experimented with a strong selection grid that includes $s = 10^{-1}$. We find that models fit with the strong selection grid have predictive accuracy, as measured with $R^2_{\text{LOCO}}$, that were about one percentage point higher. This is suggestive of stronger selection than has previously been estimated using constrained grids (see S1 Text Table 2). For this strong selection grid, we estimate average selection coefficients for the CADD 6% model of $\bar{s} = 0.042$ for CEU, 0.032 for CHB, and 0.044 for YRI. Thus, the predefined selection coefficient grid affects ultimate estimates of the DFE and average selection coefficient.

However, we find indications of model non-identifiability across the strong selection grid runs. First, estimates of the DFE with the strong grid are bimodal (S1 Fig). For example, under the CADD 6% strong selection grid model, new mutations are estimated to have a selection coefficients of $s = 10^{-1}$ with 43% chance, $s = 10^{-2}$ with 7.8% chance, and $s = 10^{-3}$ with 41% chance in the YRI samples. We propose that one mechanism for this non-identifiability is that very deleterious mutations lead to larger whole-genome reductions in diversity, which are difficult to distinguish from a smaller drift effective population size (i.e. the $\pi_0$ parameter). One

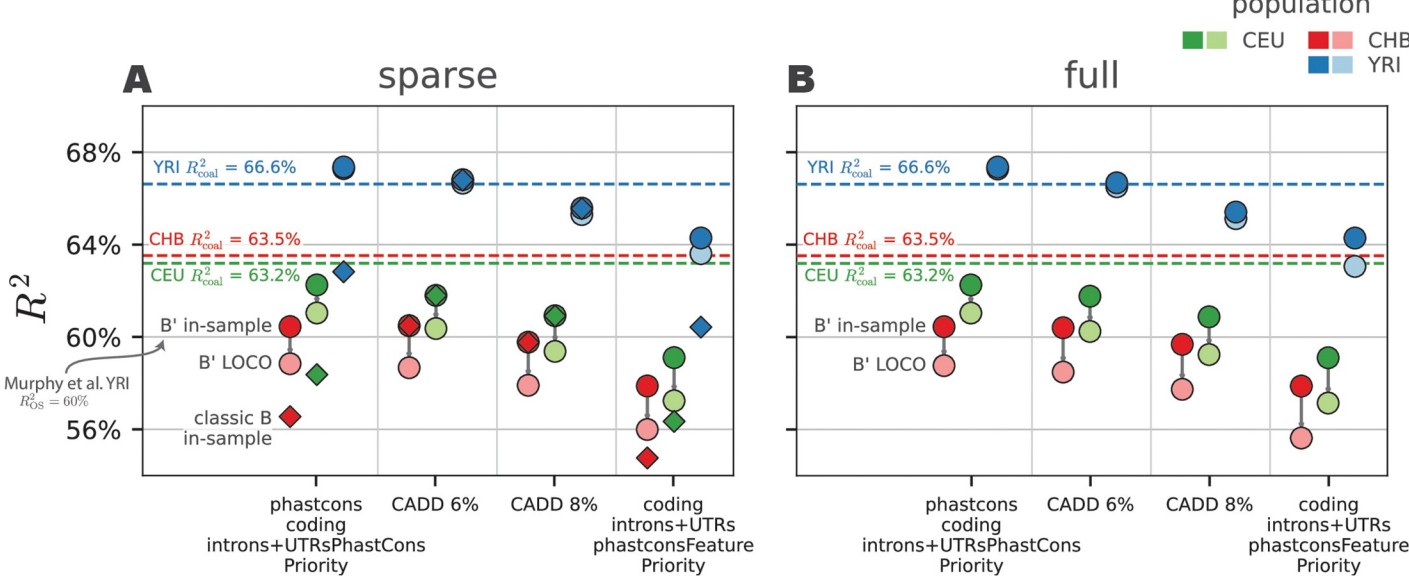

**Fig 3. The distribution of fitness effects of new mutations estimates for YRI reference samples.** (A) The DFEs using sparse (left column) and full-coverage (right column) tracks, across different annotation models (row). Color indicates the feature type. (B) The DFE of the full-coverage Feature Priority model comparing the estimates across reference population samples. Although this model fit the data less well than alternatives, its results are more interpretable.

way to test this hypothesis is to look to see if there is a systematic positive relationship across models between average selection coefficient and $\pi_0$, which is includes the drift-effective population size $N_e$. We find this is the case for all of our CADD 6% models. Across all reference samples, average selection was about 7.1 times larger using the strong selection grid, and $\pi_0$ was 5.6% higher (see S2 Fig). There was no similar consistent change in mutation rate estimates among reference samples. In the CADD 6% model, genome-wide average reduction factor $\bar{B}$ was ≈6.1% lower in the default versus constrained grid. Overall, this suggests that the linked selection signal alone cannot differentiate very strong selection from a slightly smaller drift-effective population size.

Given that it is debated how strongly demography impacts the deleterious mutation load [62–66], we were curious how consistent our DFE estimates are across samples from different reference populations. Overall, we find DFE estimates are relatively stable across samples from different reference populations and annotation models (S1 Text Section 6). Only in our Feature Priority model (Fig 3B, top row) do we see a slightly different DFE estimate for coding sequences between YRI and CEU/CHB samples, but this could be due to the poorer fit this model has to data.

Although the Feature Priority model fits the data less well than alternative models, its DFE estimates are more interpretable. We find that our B' method estimates a bimodal DFE for coding sequences for the Feature Priority model, with a large mass placed on $10^{-3} \leq s \leq 10^{-2}$ and another on the neutral class $s = 10^{-8}$. This is expected, given that the synonymous and non-synonymous sites that constitute coding sequences are under vastly different levels of constraint and are lumped together in our annotation class. Moreover, features expected to be only weakly constrained such as introns and UTRs have the bulk of DFE mass on the neutral class, with a small but significant amount of mass (≈ 3%) placed on $s = 10^{-2}$. As expected, the DFE for PhastCons regions (which in this model correspond to highly-conserved non-coding elements) suggests it is under strong selective constraint; however, we note that block jackknife-based uncertainty estimates suggest the model is uncertain whether there is some mass on the neutral class. Finally, we highlight one result from our PhastCons Priority annotation model (Fig 4A bottom row): the DFE estimate for coding sequences excluding PhastCons regions is estimated as neutral. This too is expected; the selection signal in coding regions is absorbed by the PhastCons feature, leaving only conditionally neutral sites.

**Estimates of the deleterious mutation rate are sensitive to model choice.** Prior work on genome-wide inference using the classic BGS model fit the patterns of diversity well, but led to unusually high estimates of the mutation rate [14]. This led to the hypothesis that these models could be absorbing the signal of positive selection [67], though other work has found a limited role for hitchhiking at amino acid substitutions [16, 68, 69]. While our simulation results suggest estimates of the mutation rate from linked selection models are biased, we still check for rough agreement with pedigree-based estimates [70, 71]. We find across all populations, our mutation rate estimates from CADD-based models are roughly consistent with pedigree-based estimates (Fig 5A), consistent with recent work [16]. Our full-track CADD 6% model estimates the mutation rate as $\hat{\mu} = 1.56 \times 10^{-8}$ for YRI, $1.64 \times 10^{-8}$ for CEU, and $1.60 \times 10^{-8}$ for CHB reference samples (S1 Text Section 3.6). As expected, the sparse-track CADD model mutation rate estimates are nearly identical between the B' and classic BGS methods (Fig 5A top row).

However, mutation rate estimates for feature-based annotation models do not agree with pedigree-based estimates. First, mutation rate estimates under from classic BGS theory are an order of magnitude below the expected range (Fig 5A top row). We observe similar behavior when we use the classic BGS model to fit full-coverage annotation models (Fig 5A bottom row). This behavior is consistent with classic BGS theory being unable to fit the DFE to

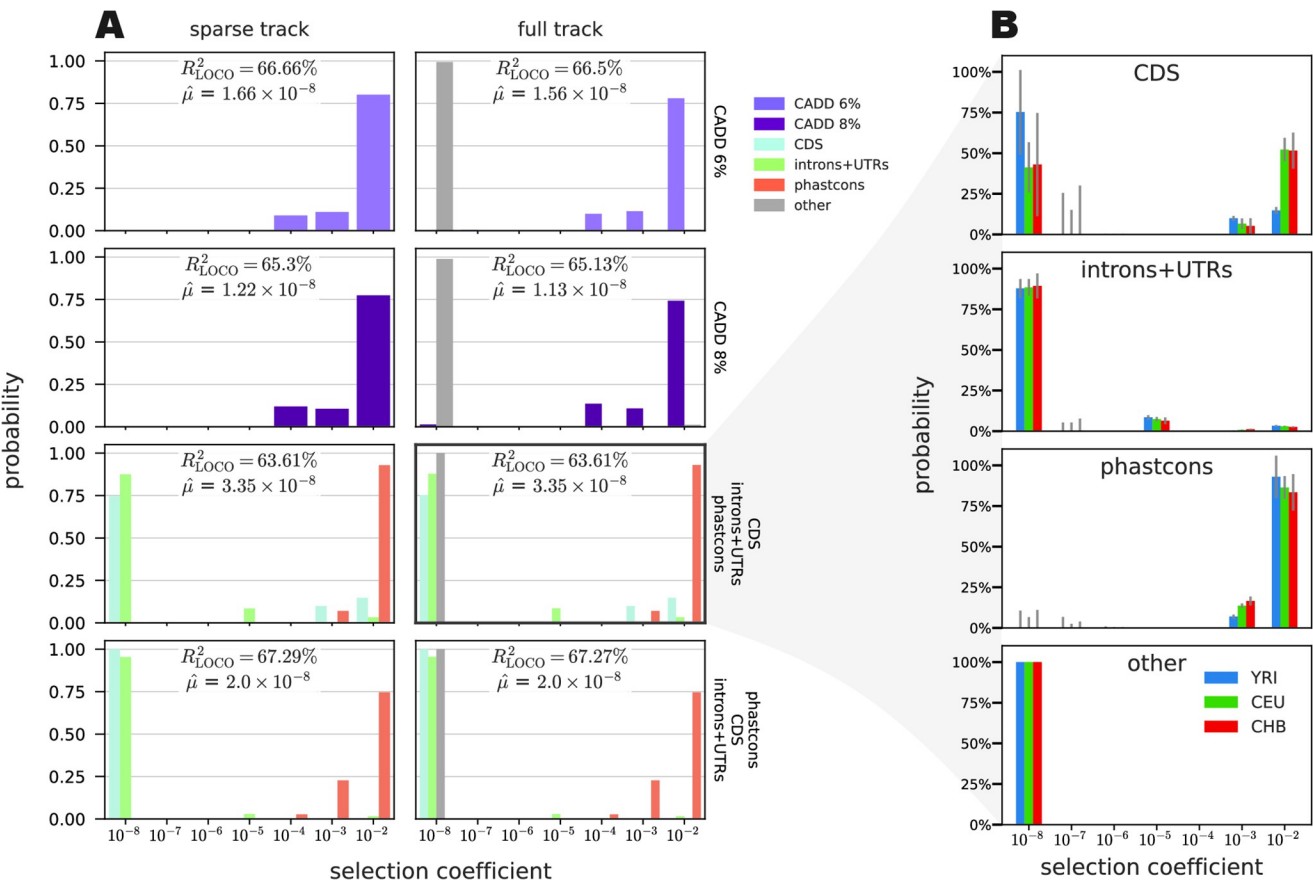

**Fig 4. The $R^2$ estimates for sparse (A) and full (B) models, for all samples (colors) fit at the megabase-scale.** Round points are our B' method and diamonds are the classic BGS (we exclude classic BGS in the full track subfigure, since these all fit very poorly). Lighter color round points are the out-sample $R^2_{\text{LOCO}}$ estimates for our B' method, and arrows show the decline in goodness-of-fit due to in-sample overfitting (out-sample $R^2_{\text{LOCO}}$ were not calculated for classic B values due to computational costs). The horizontal dashed lines are the $R^2_{\text{drift}}$ expected when the residual variance is given by the theoretic variance in coalescence times due to drift alone.

features under weak constraint (e.g. introns, UTRs, and the "other" feature), and thus must compensate by estimating too low a mutation rate.

Second, we noticed that across all populations and sparse and full tracks, the CADD 6% model consistently led to slightly higher mutation rates than the CADD 8% model (Fig 5A bottom row; S1 Text Section 3.6). This same pattern was observed in Murphy et al. [16] (Appendix 1, Fig 16). This behavior suggests a non-identifiability issue between higher per-basepair mutation rates and annotation tracks that contain more conserved sequence. This is expected from theory, since both classic BGS and SC16 models only depend on mutation rate through the compound parameter $\mu L$, where $L$ is the length of the conserved segment. Even though our method is much more robust to the inclusion of non-conserved regions like introns, we still observe this non-identifiability issue.

Finally, we note that mutation rate estimates from the Feature Priority model are themselves too high ($\widehat{\mu} \approx 3 \times 10^{-8}$), reminiscent of the high mutation rate estimates found under McVicker et al.'s model. While both our and Murphy et al.'s CADD and PhastCons-based models alleviate this issue, it is worth considering why this could occur. We can potentially gain some insight from comparing the estimated mutation rates from our Feature and

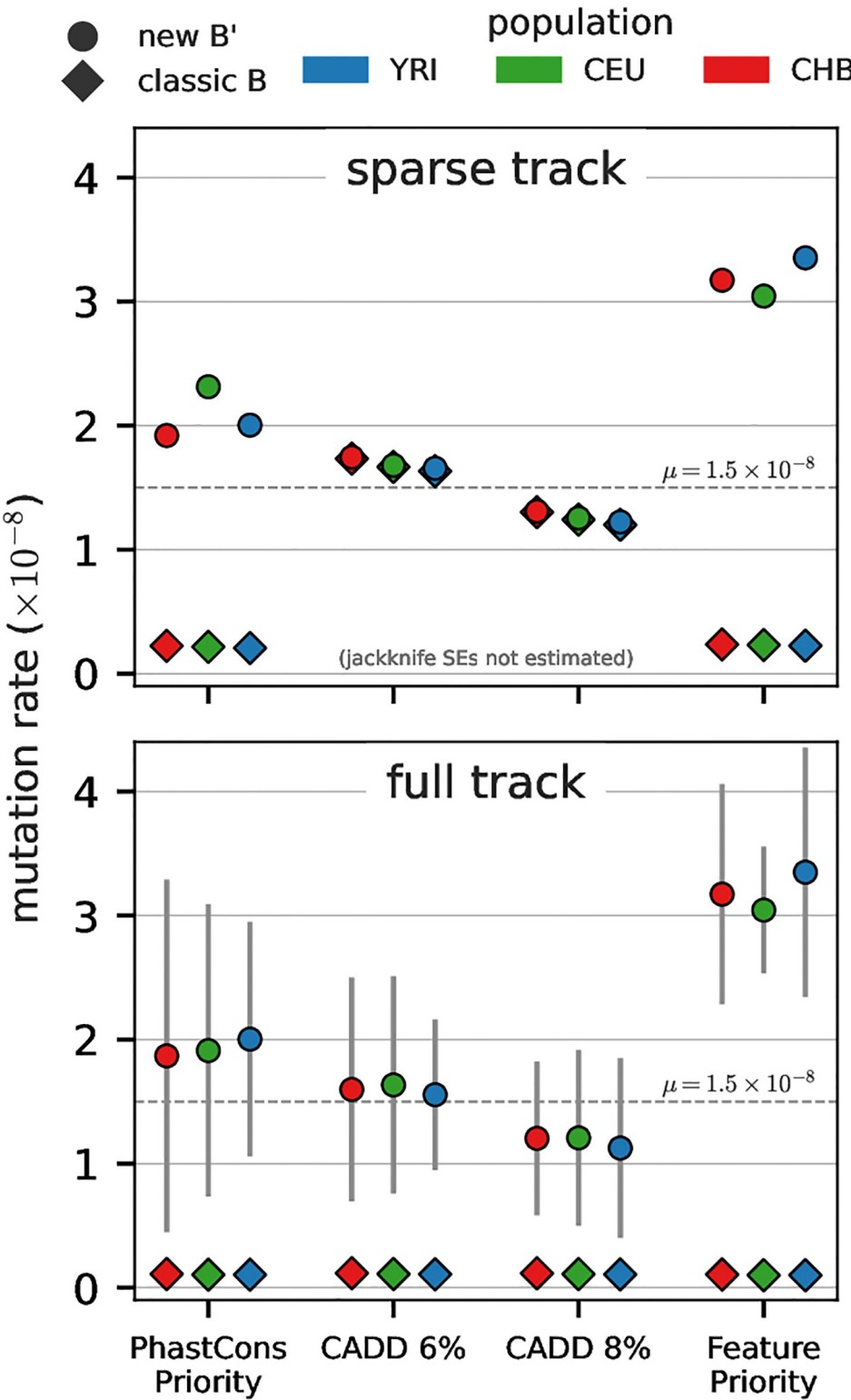

**Fig 5. Mutation rate estimates across the sparse (top row) and full-coverage tracks (bottom row) models, for the new B' (circles) and classic BGS (diamonds) methods.** Estimates of the mutation rate are consistent between classic BGS and B' methods for sparse tracks CADD models (overlapping diamonds and circles, top row). Overall, mutation rate estimates are sensitive to the underlying annotation model.

PhastCons Priority annotation models, which each contain the exact same number of feature basepairs, but whose composition varies based on the priority of overlapping features. That one of these models is our best-fitting model and the other our worst indicates that model fits are sensitive to feature classes which themselves have heterogeneous DFEs. CADD-based models fit better in part due to their fine-scale resolution of selective effects across the genome. While ideally we would fit a CADD model with different features corresponding to the different percentiles of pathogenicity, these features are on the basepair scale and thus too memory-intensive for our method to currently accommodate.

**Despite close fit, residual purifying selection signal remains.** Comparing predicted against observed diversity along chromosomes, we find a close correspondence consistent with the high $R^2_{\text{LOCO}}$ (Fig 6A). Once scaled by the genome-wide average, predicted and observed diversity levels across the genome differ little across samples from reference populations. Given that the CEU and CHB samples are from bottlenecked out-of-Africa populations and their mutation rate and DFE estimates are similar, this is an empirical demonstration that our model is fairly robust to violations of the constant population size assumption of the

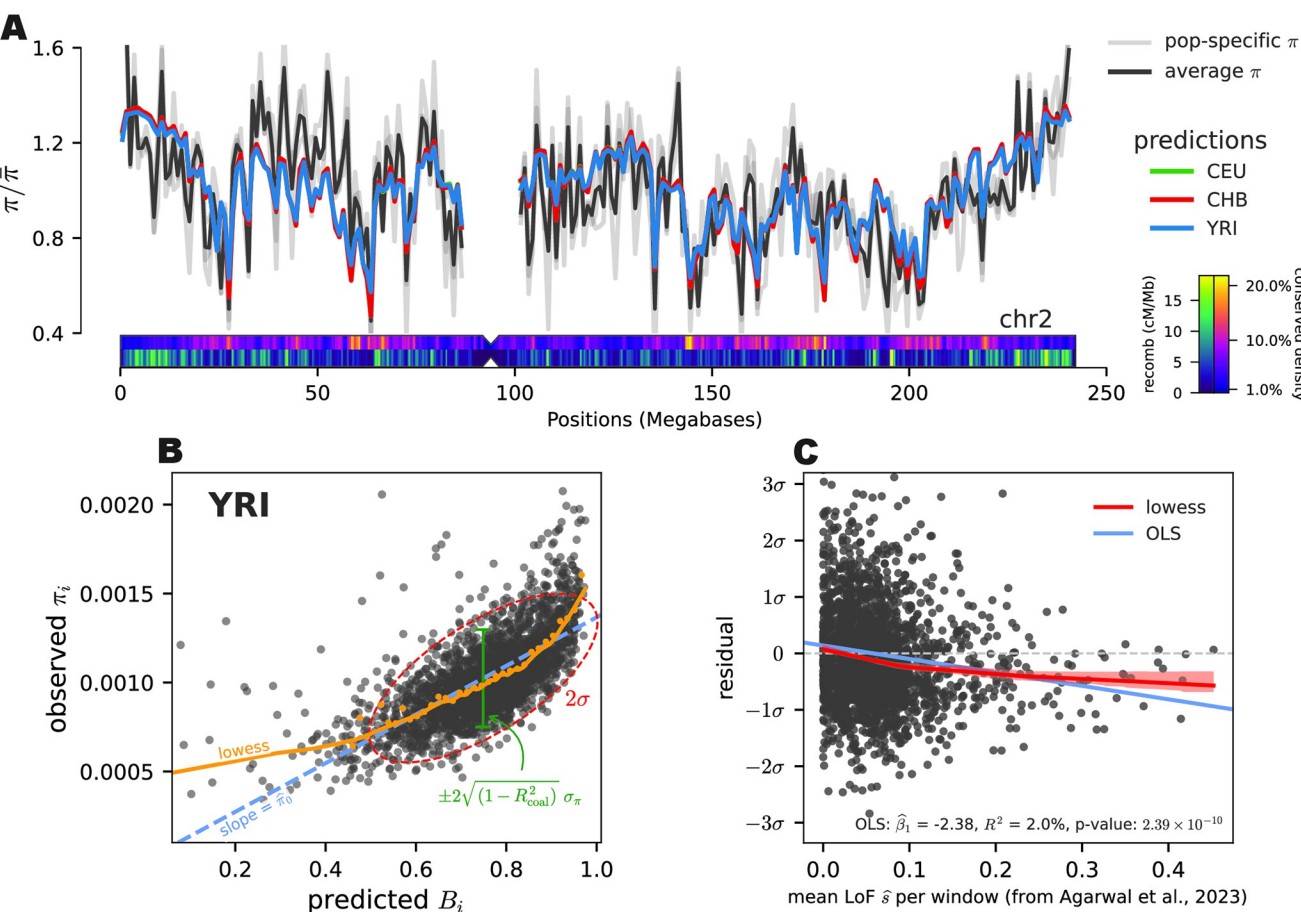

**Fig 6.** (A) Observed and predicted diversity of the B' model fit with the CADD 6% full-track annotation. Once scaled by average diversity, predicted diversity for populations (colored lines) differs little across populations, and closely matches observed diversity within each population (light gray lines). Additionally, we show summaries of CADD density and recombination rate along the chromosome below. (B) Predicted $B$ and observed $\pi$ for each window. The red dashed line indicates the observed 2 standard deviation ellipsoid, which has nearly the same width as the expected by $R^2_{\text{coal}}$, indicating the residual variance is close to theoretic expectations. The yellow points are binned means, and the yellow line is the lowess curve through predicted and observed values. (C) CADD 6% residuals (YRI shown) plotted against the average LoF selection coefficient across genes in megabase windows (estimated by [72]).

theory (see S1 Text Section 5.7 for a comparison of the predicted B' maps across different populations).

However, we note a few large (tens of megabases) regions with systematically poorer fit (S1 Text Section 5.6). In Fig 6A we see one such region on the short arm of chromosome 2, from 30 Mbp to 60 Mbp. Interestingly, predicted diversity closely follows the peaks and troughs of this region, however, predicted diversity is lower than observed. We note that a small region within this stretch had been found by a genome-wide scan for associative overdominance [73]. We further investigate this by inspecting whether observations are systematically different from predictions. We confirm a finding of Murphy et al. [16] that regions predicted to experience little reduction in diversity due to background selection (i.e. $B \approx 1$) have higher diversity than predicted (Fig 6B, orange line). Murphy et al. [16] suggested that this could reflect ancient introgression between archaic humans and ancestors of contemporary humans. Despite the prediction error in this region, the variance around observed and predicted diversity levels falls very close to what we would expect under the theoretic coalescent-noise-only expectation ($R^2_{\mathrm{coal}}$).

As DFE heterogeneity within a class of sites may be poorly fit by our model, we looked for unaccounted selection in our model residuals. First, we inspected whether there was a relationship between the fraction of CADD 2% and 6% basepairs and the residual across megabase windows (S3 Fig), finding a negative significant relationship in both cases. CADD 2% was used in this case to search for a residual signal from highly-constrained regions. Moreover, our model over-predicted diversity in windows containing more CADD 2% basepairs than CADD 6%, consistent with heterogeneity in site pathogenicity being poorly fit by our model. However, the total residual variance explained is $R^2 = 0.3\%$ and $R^2 = 0.9\%$ for the CADD 2% and 6% tracks respectively, suggesting only a modest amount of selection signal remains within the CADD annotations. There was no relationship between residuals and recombination rate (S4 Fig); we note predicted $B$ values per megabase window are strongly correlated with CADD 6% and recombination rate as expected by theory (S5 and S6 Figs).

Since our method does not include the possible effects of linked positive selection, we might expect windows containing hard or soft sweeps would have systematically lower diversity levels than predicted. Using the locations of soft and hard sweeps detected using a machine learning approach [74, 75], we tested whether the residuals of the CADD 6% model containing sweeps were systematically different than those not containing sweeps. We find no significant difference between the magnitude of residuals of windows containing sweeps versus those that do not (S7 Fig; Kolmogorov–Smirnov p-value = 0.71). The same was true if we looked at hard or soft sweeps individually as a class.

We further tested for remaining selection signal in our CADD 6% model residuals by using gene-specific estimates of the fitness cost of loss-of-function (LoF) mutations from Agarwal et al. [72]. These estimates are based on an Approximate Bayesian Computation approach that estimates the posterior distribution over LoF fitness costs from the observed dearth of LoF mutations per gene, and thus is an independent approach to assess the strength of purifying selection. We averaged the estimated LoF fitness costs across genes for each of our megabase windows, and plotted our residuals against these average LoF fitness costs. Contrary to the weak CADD residual signal described above, we find evidence of a fairly strong relationship between our residuals and average LoF fitness cost (Fig 6C; $R^2 = 2.1\%$, p-value $1.27 \times 10^{-10}$). In other words, roughly 2% of the variance in these residuals is explained by the average fitness costs of LoF mutations in the window. Consequently, our model over-predicts diversity by about $\sigma/_2$ or more in windows harboring the top 1.7% most LoF-intolerant genes.

**Predicted substitution rates indicate potential model misspecification.** Since our B'
method also predicts deleterious substitution rates $(\widehat{R})$ for each feature class, it allows for
another check of model sufficiency by comparing the predicted substitution rates to observed
levels of divergence. We estimated sequence divergence on the human lineage using a multiple
alignment of five primates for each feature in our feature-based models (Methods Section Sub-
stitution rate prediction and divergence estimates). We compared these to the predicted sub-
stitution rates per feature, averaging over all segments in the genome. Since our simulations
show that mutation rate estimates can be biased, we predicted substitution rates under a fixed
mutation rate of $\mu = 1.5 \times 10^{-8}$. Fixing the mutation rate also allows us to more easily compare
the predictions across our feature-based models. Unfortunately, a careful comparison between
our predictions and observed divergence rates is hindered by considerable uncertainty in gen-
eration times, heterogeneity in the functional constraint across genes, and the human-chim-
panzee divergence time. We assume a generation time of 28 years [76], and calculate the
sequence divergence implied by our predicted substitution rates over a range of divergence
times, from 6 Mya to 12 Mya [77–80].

We find that predicted substitution rates are qualitatively consistent with the observed
divergence along the human lineage for all features except the PhastCons regions (Fig 7). As
expected, the predicted substitution rates in features under reduced selective constraint
(introns and UTRs, and the "other" feature) are very close to the mutation rate. Throughout,
we report our substitution rates as a percent relative to the total mutation rate, $\mu$ (here fixed to
$1.5 \times 10^{-8}$). In our Feature Priority model, coding sequences are predicted to have a substitution
rate of 41.20% of the mutation rate, introns and UTRs 94.71%, PhastCons regions 0%, and the
"other" feature 99.98%. For comparison, the substitution rates along the human lineage (as a
proportion to the substitution rate in putatively neutral regions) are 74.15% in UTRs, 92.44%
in introns, 50.96% in coding sequences, and 49.56% in PhastCons regions. The large discrep-
ancy between predicted and observed PhastCons substitution rates is driven by our DFE esti-
mates suggesting that the bulk of mass is on selection coefficients greater than $10^{-3}$, which
have no chance of fixation in a population of $N_e \approx 10,000$. We note that our DFE estimates
are qualitatively similar to those inferred using the classic BGS model, so the disagreement
between observed divergence and predicted substitution rates could indicate a potential model
misspecification problem.

**Possible signal of selective interference.** Given the prediction error for substitution rates
in highly-conserved regions and that simulation indicates that $B(x)$ is more accurately pre-
dicted when we use local rescaling, we modified our composite-likelihood method so that it
can be run a second time, on B' maps locally rescaled by the predicted $\widehat{B}(x)$ from the initial fit.
Intuitively, this is based on the notion that if a neutral allele experiences a fitness-effective pop-
ulation size of $B(x)N$, so too should a selected allele, and this should be considered in how the
SC16 equations are solved. This is an approximation to selective interference, since interfer-
ence acts to lowers the effective population size in other regions [26, 27, 50].

There are five important but tentative results to draw from this analysis. First, estimated
mutation rates are in general higher. Under the CADD 6% model, they are $\widehat{\mu} = 6 - 7.3 \times 10^{-8}$ across populations; for the PhastCons Priority model, they reach the upper limit of our
optimization boundary of $\mu = 8 \times 10^{-8}$ (S1 Text Section 5.1). Second, all of our leave-one-chro-
mosome-out $R^2_{\text{LOCO}}$ are about one percentage point higher than the unrescaled model. Third,
the DFE estimates for both CADD 6% and PhastCons regions in the PhastCons Priority model
is now U-shaped (S8 Fig), with 70–77% of mass being placed on a weakly deleterious class,
$s = 10^-5$. Interestingly, this is the first of all of our models where such an appreciable mass has
been placed on a midpoint in our selection coefficient grid; in all other cases, non-strongly

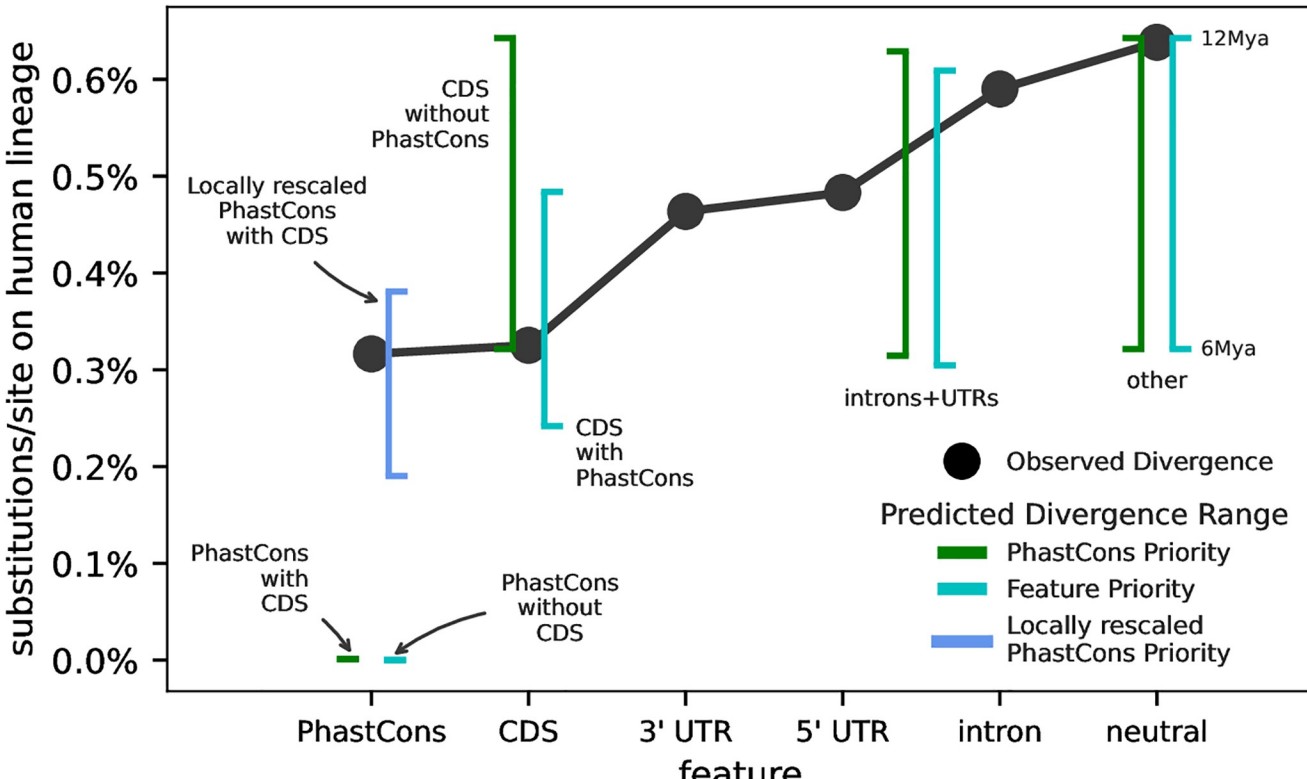

**Fig 7. The divergence implied from predicted substitution rates under the B' model versus observed divergence along the human lineage.** Black points are the PhyloFit divergence rate estimates per feature (on x-axis). Line ranges are the implied divergences across a range of human-chimpanzee divergence times of 6–12 Mya (using a generation time of 28 years). We show the predicted divergences for our Feature (turquoise) and PhastCons priority (green) annotation models. Additionally, we show the predicted PhastCons region divergences when local rescaling is applied (blue; we omit other locally rescaled predictions since these to not differ substantially).

deleterious estimates were neutral ($s = 10^{-8}$). Fourth, predicted pairwise diversity is nearly identical to our original, non-rescaled fit (see S9 Fig). Finally, since local rescaling increases the DFE mass over $s < 10^{-3}$, mutations in PhastCons regions now have the possibility of fixation. We find that local rescaling the PhastCons Priority, leads the predicted substitution rates in PhastCons regions to be much closer to observed levels (blue line, Fig 6).

Finally, we note an important caveat about this analysis. Since local rescaling is done using the first round of maximum likelihood estimates, there is some possibility of statistical "double dipping", since the $B(x)$ at this position includes the contribution of the focal segment that is being rescaled, and it has already been included in the initial fit that lead produced the predicted $B(x)$ map. Ideally, one would exclude this segment's contribution to $B(x)$; however, this is computationally unfeasible. However, two observations indicate our findings here are relatively robust despite this limitation. First, the results do not change based on whether the $B(x)$ is averaged at the 1 kbp level or at the megabase scale; for the latter, a single segment makes little contribution to the average. Second, we investigated the extent to which local rescaling modified the B' maps across selection parameters. We find minor differences between the locally rescaled and standard B' maps for fixed selection coefficients (i.e. before model fitting) in the nearly-neutral domain ($0.2 \leq 2N_e s \leq 2$). Additionally, the locally rescaled and standard maps are identical under strong selection ($2N_e s = 20$) as expected (S10 Fig). Moreover, the correlations between the standard and locally rescaled B' maps across the genome are high (100%

for $2N_e s = 20$, 96.5% for $2N_e s = 2$, and 60.21% for $2N_e s = 0.2$). The overall realized effect of local rescaling is to just alter how deep the "U" is in the relationship between the reduction factor $B$ and the selection coefficient (S11 Fig).

Overall, this suggests that models of the signal of linked selection are worryingly sensitive to the theoretic B' values in the $2Ns \leq 1$ domain. The fact that predicted diversity differs little between standard and locally rescaled B' methods indicates there may not be enough information in pairwise diversity alone to differentiate when interference is occurring or the causes of fitness variance along the genome. Moreover, local rescaling turns out to only slightly alter the B' maps, yet significantly modifies the DFE estimates. This brings the deleterious substitution rate in agreement with observations (since this is predicted with a fixed $\mu = 1.5 \times 10^{-8}$; however, the maximum likelihood estimate of mutation is implausibly high. This suggests either the local rescaling approximation to interference is not suitable (though our chromosome-wide simulations show locally rescaled B' maps are close to the reductions observed from simulations), or that the deleterious mutations-only model does not adequately describe the processes generating fitness variance.

## Discussion

New mutations at functionally important regions of the genome are a major source of fitness variation in natural populations, as the vast majority of such mutations are deleterious. Purifying selection, working to remove these deleterious variants, perturbs genealogies at linked sites, creating large-scale patterns in genomic diversity. While this has been recognized for decades [8, 17], the availability of genomic data allows for methods to estimate the degree to which purifying selection shapes genomic variation and at what scale.

Accordingly, there have been a number of recent efforts to fit parametric models of linked selection to polymorphism and divergence data in Drosophila [15] and humans [14, 16]. These efforts have yielded reasonable estimates of the strength of selection on new mutations as well as provided mutation rate estimates that largely agree with pedigree-based estimates. However, previous methods have relied on the canonical background selection model, which assumes that mutations are sufficiently deleterious such that they cannot fix. Consequently, statistical methods using the BGS model should only be expected to fit well when some regions are *a priori* under strong selective constraint. In reality, the relationship between neutral diversity levels and the strength of selection from purifying selection in linked regions is U-shaped, which implies there could be more uncertainty than previously appreciated in the distribution of weak and strongly deleterious mutations.

In this work, we developed and fit a different class of linked selection models based on the equilibrium fitness variance [28, 29]. Fundamentally, we model the reduction in diversity as a function of how additive fitness variance is distributed along the recombination map of the genome [28]. We fit a specific model for this fitness variance that supposes all variation is the result of selection against new additive deleterious mutations [29]. Unlike classic background selection theory [8, 11, 13, 17], the SC16 model explains equilibrium fitness variance across all selection coefficients by jointly predicting another central quantity in evolutionary genomics: the substitution rate of deleterious alleles.

Our method has at least four improvements over previous whole-genome linked selection methods based on the BGS model. First, our model leads to better fits to data than those based on classic BGS, as measured by predicted out-of-sample diversity. Second, unlike BGS-based methods, our model is capable of fitting weak selection. When regions under weak or little selective constraint are included in methods using classic BGS, parameter estimates can become severely biased. By contrast, we have demonstrated via simulation that our method

can estimate the strength of selection even for weakly constrained features (e.g. introns and UTRs), as well as remaining unannotated regions of the genome. Third, fitting our model produces a simultaneous prediction of substitution rates, which can be compared to observed divergence rates. Finally, the effect of selective interference can be approximated by locally rescaling the B' maps, which our forward simulations show reduce prediction error of genome-wide diversity levels.

Even though our model is able to fit weak selection, our initial estimates of mutation rate and DFEs were consistent with prior work [16]. This, at first glance, suggests further confirmation that strong purifying selection is the dominant mode of linked selection in the genome. However, we find that predicted substitution rates for highly-conserved PhastCons features disagree with observed rates of divergence along the human lineage. This disparity between observed divergence and predicted substitution rates is likely a consequence of our DFE estimate for PhastCons regions containing little mass over weakly deleterious and neutral selection coefficients that would have some possibility of fixation—a characteristic of DFE estimates from other work too [16].

Our simulation results reveal another possible source of disagreement: in the weak selection domain of $2N_e s \approx 1$ there is an appreciable level of disagreement between theory and simulation. We hypothesize that this could be because as $2N_e s$ approaches 1, a segment under selective constraint experiences a local fitness-effective population size of $B(x)N$, and not just $N$. This local fitness-effective population size is induced by selection at other segments that is not being taken into account by classic BGS theory or our standard SC16 model. When we experimented by fitting our model and then using the predicted reduction map to locally rescale $N_e$ to the fitness-effective population size $N_f = \widehat{B}(x)$, we found the disagreement between predicted and observed substitution rates disappears. This is expected since locally rescaled DFE estimates have an appreciable mass on weakly deleterious selection coefficients, contrary to the standard fits.

This pattern is consistent with a scenario where the same selection processes that reduce diversity over long stretches along the chromosome also decrease the efficacy of purifying selection. This idea has been proposed before in an extension to the McDonald–Kreitman test that accounts for how background selection can bias estimates of the proportion of adaptive substitutions [81]. While our simulation results indicate local rescaling reduces error in the weak selection domain, it is worth noting some caveats about this approach. First, local rescaling is only an approximation to selective interference; as our simulations show, this approximation reduces error in the predicted reduction $B(x)$, but this may not fully account for how negative linkage disequilibrium builds up and reduces fitness variance. A benefit of the SC16 approach is that the equations can be solved with a locally rescaled effective population that approximates this process. Second, there is the possibility of circularity, since a preliminary fit must be made to estimate $B(x)$, which is then used to re-solve the SC16 equations with a local fitness-effective population size.

Despite these caveats, the local rescaling approach suggests selective interference could alter inferences about the DFE and bring predicted substitution rates into agreement with observed divergence rates. However, these results also demonstrate that parameter estimates are extremely sensitive to how accurate the theoretic B' maps are in this domain. Moreover, we find that predicted diversity differs little across DFE and mutation rate estimates, suggesting there may be limited information in pairwise diversity to differentiate between models, thus inclusion of allele or haplotype frequency information might be informative in the future. Still, our mutation and DFE estimates are relatively stable across reference populations, suggesting these estimates are not too noisy, though they may be biased due to model misspecification.

While local rescaling brings substitution rates into agreement, it also re-introduces a similar problem found by Murphy et al. [14]: the estimated mutation rate is too high. While our mutation rate includes point mutations as well as all other forms of deleterious variation (e.g. insertions/deletions, copy number variants, etc.), our estimations suggest exceedingly high rates of deleterious variation per generation.

Examination of the local residuals of our predicted diversity levels demonstrated no systematic effect of previously identified hard and soft selective sweeps [75]. This result echoes what has been observed in previous efforts to look at genome-wide patterns of linked selection [16, 69], and suggests that the scale of perturbation due to selection sweeps is more restricted (e.g. at the kilobase scale, [82]) than the scale at which we are modeling variation. Taken together this suggests that selective sweeps are not likely responsible for shaping the majority of variance in large-scale patterns of chromosomal variation in humans.

Given that our model is essentially parameterized by levels of fitness variance along the genome, the high estimates of mutation rate could suggest that purifying selection is not the only source of fitness variance generating the genome-wide linked selection signal in humans. Selection on polygenic traits could be another source of fitness variance, since the underlying theory suggests levels of pairwise diversity are determined by total additive fitness variance (i.e. Eq (2)). Alternatively, purifying selection could be the main source of fitness variance, but the complexity of selective interference may be poorly approximated by the local rescaling approach. Another possibility is that our assumption throughout of additive effects may lead to biases given the vast majority of deleterious mutations are partially recessive. Additionally, our approach has ignored the possibility of back mutations at sites at which there had been a prior deleterious substitution, which also can bias the predicted reduction in diversity. Future theoretic work testing the robustness to these forms of model misspecification, perhaps with realistic forward simulations of multiple modes of linked selection (e.g. [83]) are needed to fully disentangle these processes. Additionally, our DFE estimates are done using a discretized logarithmic-spaced grid following previous work [15, 16]; future work could explore continuous and parametric forms for the DFE. As we find evidence of strong selection against loss-of-function in our model residuals, it is also possible that the bulk of fitness variance *is* due to purifying selection, but our model is unable to account for strong heterogeneity in the DFE per annotation class.

Moving forward it remains a central goal to understand how the sources of fitness variation shape the striking patterns of diversity along the human genome. Our work embeds this question in the quantitative genetic framework that is more accurate and flexible than proceeding models, but there is much work yet to do to incorporate important population genetic features such as dominance effects and selective interference. Overall, the complex interplay of mutation, selection, drift, and interference may confound our understanding of selection in the human genome for some time.

## Methods

### Solving the B' equations for each segment

Our software `bgspy` [84] first calculates the equilibrium additive genetic fitness variation $\widetilde{V}_A$ and deleterious substitution rate $\widetilde{R}$ for each user-specified segments in the genome. These equilibria are calculated across grids of mutation rate weighted by the DFE $m_i$ and selection

coefficient $s_j$, by numerically solving the following system of equations,

$$N_f = N \exp\left(-V_A \frac{Q^2(m_i, s_j)}{2}\right) \qquad \textit{effective population equation} \qquad (8)$$

$$R = \frac{4N_f U s_j}{\exp(4N_f s_j) - 1} \qquad \textit{substitution equation} \qquad (9)$$

where,

$$V_A = (U - 2R)s_j \qquad \textit{fitness variance equation} \qquad (10)$$

$$Q^2(m_i, s_j) = \frac{2}{(1-Z)(2-(2-M)Z)} \qquad \textit{linkage inflation factor} \qquad (11)$$

$$Z = 1 - \frac{Us_j}{U - M} \qquad \textit{variance decay rate} \qquad (12)$$

and $U = m_i L$ and $M = r_{\mathrm{BP}} L$ are the total mutation and recombination rates in the segment. A detailed derivation of these equations can be found in S1 Text Section 1. The recombination rate in a segment is determined by a user-supplied recombination map.

## Calculating the reduction maps

Our method uses the pre-computed equilibria $\widetilde{V}_{A,g}$ for each segment $g$ (specified by the particular annotation model) to calculate the reduction map $B(x; m_i, s_j)$ at positions $x$ across the parameter grids described above. Since we assume multiplicative fitness, the reduction is the product of each segment's contribution accounting for the recombination is,

$$B(x; m_i, s_j) = \exp\left(-\frac{1}{2} \sum_{g \in G} \widetilde{V}_{A,g}(m_i, s_j) Q_g^2(m_i, s_j, r_{x,g})\right) \qquad (13)$$

where $r_{x,g}$ is the recombination fraction between the focal site and segment $g$. Here, $Q^2(m_i, s_j, r_{x,g})$ is given by Eq (3) squared. A separate reduction map is calculated for all features $G$ within a specific feature type. We calculate B' calculate for $\log_{10}$-spaced grids over $10^{-1} \leq s \leq 10^{-8}$ and $10^{-11} \leq m \leq 10^{-7}$, in 10kb increments across the genome.

## Composite likelihood and optimization

Following previous approaches [14–16], we use a composite likelihood approach to fit our negative selection model. Per-basepair allele count data (described below) is summarized into the number of same and different pairwise differences per window. All of our primary models were fit with megabase windows, since previous work has found the strongest selection signal at this scale (we confirm this with one CADD 6% fit at the 100 kbp scale).

Our binomial likelihood models the number of different pairwise comparisons observed per window given the total number of pairwise comparisons. The binomial probability for window $b$ is $\bar{\pi}(b; \Psi) = \pi_0 \bar{B}(b; \mu, \mathbf{W})$, where bars indicate averages over some bin width. The free parameters $\Psi = \{\pi_0, \mu, \mathbf{W}\}$ are the expected diversity in the absence of selection ($\pi_0$), the mutation rate ($\mu$), and the distribution of fitness effects for the discretized selection grid and $K$

features (**W**). The reduction at position $x$ is then,

$$\log\left(B(x; \mu, \mathbf{W})\right) = -\frac{1}{2} \sum_{g \in G} \sum_{j=1}^{n_s} V_g\left(\mu W_{j,k(g)}, s_j\right) Q_g^2(m_i, s_j, r_{x,g}). \tag{14}$$

See S1 Text Sections 2.5 and 3.11 for more details.

Our method uses two tricks to improve optimization over the mutation and DFE parameters. First, we find our pre-computed $B(x;\mu, \mathbf{W})$ reduction maps (described in the previous section) are exponential over columns of $\mu\mathbf{W}$, which allows for optimization over this smooth function rather than the grid. Second, we use softmax to convert constrained optimization over the DFE columns (which must sum to one) to unconstrained. We tested multiple different optimization routines, finding that BOBYQA outperformed alternatives [85, 86]. We inspected and confirmed convergence with diagnostic plots finding stable optima across 10,000 random starts (see S1 Text Section 3.10). We assessed model fit using out-sample predictive error, calculated by leaving out a whole chromosome during fitting and predicting its diversity. To calculate uncertainty, we used a block jackknife approach in 10 Mbp windows (S1 Text Section 3.13). All model fits, analyses, and produced data are available on Dryad [87].

### Human population genomic data

Our analyses was conducted on the Yoruba (YRI), European (CEU), and Han Chinese (CHB) reference sample individuals from the high-coverage 1000 Human Genomes data aligned to GRCh38/hg38 [88]. Since nucleotide diversity is a ratio estimator, it can be biased when subtly different filtering criteria are applied to variant and invariant sites. To prevent this, we conducted our analyses on Genomic VCF (gVCF) files that contain genotype calls for both variant and invariant sites [89]. Then, we apply the same genotype filtering criteria to all called sites (S1 Text Section 3.2). We also applied sequence accessibility masks that containing only non-repeat, non-centromeric sequence that passed the 1000 Genomes strict filter ([90]; S1 Section 3.3). Since our theory only considers the indirect effects of linked selection on a site, we additionally masked sites that are likely under direct selective constraint (see S1 Text Section 3.4). Finally, for every basepair passing these filtering and masking criteria, we counted the number of reference and alternative allele counts (excluding all multiallelic, indel, and CNV variants).

For all of our main analyses, we used the recombination map from Halldorsson et al. [91] estimated from a trio-based design to avoid circularity that could occur by using LD-based maps. We use Ensembl gene annotation [92], a special CADD Score dataset with McVicker B scores removed (to avoid circularity; [93, 94]), and PhastCons regions [2]. We did not account for mutation rate heterogeneity along the genome, since this would require using divergence-based estimates of local mutation rates that would introduce circularity when we predict divergence rates under our model.

### Forward simulations

We conducted forward simulations of negative selection on whole human chromosomes to validate our method at two stages. First, we simulated negative selection on chromosome 10 using a realistic recombination map and putatively conserved features to confirm that our classic B and new B' maps matched the average simulation reduction map across mutation and selection parameters. Second, we evaluated our composite likelihood method by simulating negative selection on the first five human chromosomes, across grids of fixed mutation and selection parameters. We then combined these into a synthetic genome, and overlaid mutations on the ARG. Then, we ran our likelihood methods on the resulting allele count data to

assess model accuracy. We ran additional synthetic genome simulations like these to evaluate the impact of two model violations: recessivity of deleterious mutations and expanding populations. For the latter, after 9.3 generations, we grew the population by factor of 1.004 each generation to mimic the human expansion out-of-Africa [95]. We did not simulate population bottlenecks since our analyses showed little difference between bottlenecked out-of-Africa samples (CEU and CHB) and YRI samples. More details about these and the segment simulations shown in Fig 1A–1C can be found S1 Text Section 4.

## Substitution rate prediction and divergence estimates

Substitution rates were predicted by resolving Eq (6) for the given estimated product between mutation rate and DFE weight, $w_{i,j} = \mu W_{i,j}$. We estimated the divergence along the human branch using PhyloFit [96] run on a subset of the UCSC 17-way Multiz alignments [97] consisting of humans and four other primates (*Pongo abelii*, *Pan troglodytes*, *Pan paniscus*, *Gorilla gorilla*). PhyloFit was run using the HKY85 substitution model per-feature; estimates from alternate substitution models yielded equivalent results. Further details about this process can be be found in the GitHub repository (https://github.com/vsbuffalo/bprime).

## Supporting information

**S1 Text. Supporting information.**
(PDF)

**S1 Fig. The DFE estimates for the strong selection grid (up to $s = 10^{-1}$).**
(TIF)

**S2 Fig. The maximum likelihood estimates of $\pi_0$ and average selection coefficient implied by the estimated DFE for the CADD 6% models.** Diamonds indicate estimates under the strong selection grid (up to $s = 10^{-1}$) and circles indicate estimates under the default grid (up to $s = 10^{-2}$).
(TIF)

**S3 Fig. Residuals from the CADD 6% sparse track plotted against the fraction of basepairs in a window annotated by a CADD region.**
(TIF)

**S4 Fig. Residual plotted against the average recombination rate in megabase window (for CADD6 model).**
(TIF)

**S5 Fig. Predicted $B$ per Mb window plotted against fraction of window overlapping a CADD 6% element, colored by recombination rate.**
(TIF)

**S6 Fig. Predicted $B$ per Mb window plotted against average rec. rate per Mb window.**
(TIF)

**S7 Fig. The distributions of residuals in windows containing hard or soft sweeps (blue) and not containing sweeps (orange) found by [75].**
(TIF)

**S8 Fig. The DFE for YRI samples with local rescaling.** The maximum likelihood mutation rate estimate for this is $\hat{\mu} = 8 \times 10^{-8}$, which is the upper boundary of the range used during

optimization.
(TIF)

**S9 Fig. Chromosome 2 predictions on the YRI samples, with locally rescaled model fits.** The observed data is the dark gray line, and the normal MLE for the PhastCons Priority model is the blue line. The locally rescaled predictions are the green line. The dashed red line are the prediction using the standard B' map (without local rescaling) and the maximum likelihood estimates from the locally rescaled fits. The large discrepancy in this shows that estimates are highly dependent on the B' map.
(TIF)

**S10 Fig. The standard (dark gray) and locally rescaled (blue) B' maps for different $\mu$ = $1.6 \times 10^{-8}$ and three different selection coefficients.** For $s = 10^{-5}$ ($2N_e s = 0.2$), locally rescaling alters the predicted reduction so that it is essentially insignificant ($B \approx 1$). For mid-strength selection $s = 10^{-4}$ ($2N_e s = 2$), there is only a very slight difference between standard and locally rescaled B' maps. Finally, for strong selection $s = 10^{-3}$ ($2N_e s = 20$), local rescaling does not change the B' maps, as expected.
(TIF)

**S11 Fig. Genome-wide average $B(x)$ values across selection coefficients for $\mu = 1.58 \times 10^{-8}$, for both the standard (blue) and locally rescaled B' (orange) maps.** This indicates that locally rescaling the B' maps only in practice changes how deep the "U" is.
(TIF)

## Acknowledgments

We would like to thank Doc Edge, Ben Good, Taylor Kessinger, Graham McVicker, Priya Moorjani, David Murphy, Rasmus Nielsen, Guy Sella, Joshua Schraiber, and Peter Sudmant for helpful discussions, and Martin Kircher for providing modified CADD tracks. We thank Brian Charlesworth, Graham Coop, Matt Hahn, Nate Pope, Enrique Santiago for comments on the manuscript.

## Author Contributions

**Conceptualization:** Vince Buffalo, Andrew D. Kern.

**Data curation:** Vince Buffalo.

**Formal analysis:** Vince Buffalo.

**Funding acquisition:** Andrew D. Kern.

**Investigation:** Vince Buffalo, Andrew D. Kern.

**Methodology:** Vince Buffalo.

**Project administration:** Vince Buffalo, Andrew D. Kern.

**Resources:** Andrew D. Kern.

**Software:** Vince Buffalo.

**Supervision:** Andrew D. Kern.

**Validation:** Vince Buffalo, Andrew D. Kern.

**Visualization:** Vince Buffalo.

**Writing – original draft:** Vince Buffalo.

**Writing – review & editing:** Vince Buffalo, Andrew D. Kern.

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
