## [Decision Letter · Decision Letter 0]

26 Oct 2023

Dear Dr Buffalo,

Thank you very much for submitting your Research Article entitled 'A Quantitative Genetic Model of Background Selection in Humans' to PLOS Genetics.

The manuscript was fully evaluated at the editorial level and by independent peer reviewers. The reviewers appreciated the attention to an important problem, but raised some concerns about the current manuscript. Based on the reviews, we will not be able to accept this version of the manuscript, but we would be eager to review a revised version. We cannot, of course, promise publication at that time.

The three reviewers (and I) are united in their enthusiasm for your study. Incorporating effects of weak selection into the background selection model and devising an improved method for quantifying background selection along the genome are important developments. The reviewers provide thoughtful feedback, the majority of which focuses on increasing the clarity of presentation. In addition to key theoretical issues, a few questions deserve special attention in a revision. How concordant are B' maps from different human populations across the genome? How can departures of the inferred distribution of fitness effects from previous estimates be explained?

If you decide to revise the manuscript for further consideration at PLOS Genetics, please aim to resubmit within the next 60 days, unless it will take extra time to address the concerns of the reviewers, in which case we would appreciate an expected resubmission date by email to plosgenetics@plos.org.

Please do not hesitate to contact us if you have any concerns or questions.

Yours sincerely,

Bret Payseur

Academic Editor

PLOS Genetics

Justin Fay

Section Editor

PLOS Genetics

Reviewer's Responses to Questions

**Comments to the Authors:**

Reviewer #1: Statement: I read an earlier version of this manuscript and provided the authors with some comments, as noted in the Acknowledgments section. However, I am in a good position to give an unbiased evaluation, since there has been no other previous collaboration between us.

E. Santiago

Here, the authors apply the model of Santiago and Caballero (2016) to explain the observed diversity and the divergence rate in humans, assuming that most selective forces shaping variation are deleterious. While this model takes into account the development of negative LD between selective variants (i.e., the Hill-Robertson interference), which leads to a reduction of the additive variance for fitness (i.e. to a reduction of the effectiveness of selection), the classical background selection model (BGS) does not predict this effect and consequently ignores the increased rate of fixation of deleterious variants when the product of the Ne and the selective effect (s) is small. This aspect is of particular interest in the human lineage, which has experienced long periods of bottlenecks that have led to reduced nucleotide diversity and gene loss, the latter presumably due to increased accumulation of mutations in non-essential genes. The authors have done a considerable amount of good work, correctly applying the theory and comparing the predictions with those of the BGS model. Simulations show the superiority of the SC16 theory in predicting all aspects of diversity and divergence, especially when selection is weak. However, some deviations from observations remain, particularly in the prediction of mutation rates.

I have found that the quantitative predictions follow the basics of the theory, but disagree on some interpretations of the results. The most important is due to the misconception that the SC16 model does not take into account for the Hill-Robertson interference, which however is believed to be accounted for by simply accumulating the effects of segments along the chromosomes. I hope that the following comments help to improve the manuscript:

L64 – I agree that the classical BGS model does not take the Hill-Robertson interference into account at all. But the sentence suggests that this is an issue related to the division of the genome into segments. It says that BGS assumes no interference between segments… but says nothing about within segments.

A second point regarding this sentence is that I have not been able to find out how long the segments are, given either by a fixed number of Mbs, or by an interval, or by a criterium. There is also little information about recombination rates, which are as important as the selection intensities for background selection models and can vary across the genome.

L180 – Several citations are given regarding the opinion that determining the rate of the Muller´s ratchet in asexual species is a difficult problem. The paper by Santiago and Caballero (2016) should also be cited here, as their Figure 4 shows the predictions of the ratchet rate by the SC16 model along with the predictions by the other methods. In general, SC16 predicts better than the other methods by simply using the same set of equations used in the manuscript. At equilibrium, there is on average one fixation per click of the ratchet (Higgs and Woodcock 1995), so predicting the fixation rate, which is implicit in SC16 method, is the same as predicting the ratchet rate. The equations in the SC16 method also connect the fixation rate (substitution rate) with the reduction of the additive variance (i.e. reduction of the effectiveness of selection due to negative LD between selected sites in the so-called the Hill-Robertson interference), making it clear that the Muller´s ratchet is the consequence of the Hill-Robertson interference in asexual species, as pointed out by Felsenstein (1974).

L267-275 – Although it is not clearly explained how the local re-scaling of the B maps is done, this paragraph suggests that the focal segment is considered twice, or its weight is somehow increased over the true contribution. Perhaps a simple simulation could help to assess the excess of effect of the re-scaling.

Also at the beginning of the paragraph, it is said that the interference between segments is not considered in the set of equations of the SC16 model. Selected sites (or segments) accumulate their effects on the linked focal site (or segment) as shown by the sums and integrals in equations (3) and (4) and other equations in the Appendix. This accumulation is also considered in the classical BGS model. Interference means LD disequilibrium between selected sites, which is also considered in the SC16 model but not in the classical BGS model. Solving the set of equations for a single segment, i.e. ignoring the existence of the rest of the genome, leads to the overprediction of Ne because there are no linked segments to accumulate their effects on the focal segment. In addition, the negative LD with selected sites on other segments, if exists, disappears but this probably contributes little to the variance because most LD occurs between tightly linked sites, most of which would be expected to be found within the same segment, unless the segment were tiny.

L-282 of the main text and L-343 of the Appendix – It is stated that V_a in eq.(5) [eq.(43) in the Appendix] is the additive “genic variance”. I understand that this means the sum of the contributions of all the selected loci as if they were independent (uncorrelated), i.e. breaking all the associations between loci and adding their contributions. If so, then this concept in the manuscript is wrong according to the SC16 model. Fisher’s fundamental theorem states that the rate of increase of fitness per generation is equal to the additive variance for fitness. This applies exactly in selection models without dominance because it is a definition of the additive genetic variance (Li 1967). Of course it refers only to the variance that is visible to natural selection, so the variance hidden behind negative linkage disequilibrium is not accounted for in V_a. Fisher’s equation is in the center of the SC16 model, which means that the variance hidden behind LD is ignored when predicting the rate of fixation and the effective population size. However, the hidden variability contributes to the nucleotide diversity in an amount that is ignored by the model, leading to an overprediction of the mutation rate when trying to explain the observed diversity in the range of parameters where the negative LD occurs, mainly when Nes<5 approx. , as demonstrated by solving the set of equations of the SC16 model and also shown by simulations in Figure 1D (top).

L284 – Is D a covariance of gene frequencies? It is probably not necessary to include this term in the main text, since it is only used here.

L297, L305, L321, L357, L494 – Some references to figures or parts of figures are incorrect.

L638 , L676– Certainly, local rescaling, which is a tactic to account for the cumulative contribution of selected sites located beyond the limits of the segment, has little to do with Hill-Robertson interference.

L690 , L698 – Of course, like neutral alleles any selected allele will experience the increase in drift caused by selection acting on linked sites. However, if selection is strong (i.e. Nes>>1), this will not significantly affect the evolution of their frequencies. On the contrary, weakly selected alleles may become fixed as a consequence of the Hill-Robertson interference.

L729-740 – Trying to explain the overestimation of mutations rates, several factors not considered in the model are simply mentioned without discussion of how they might affect the estimates. Two of these seem obvious. The first, but probably less important, is that the model predicts well the observed additive fitness variance, but ignores the hidden variance, which however contributes to genomic diversity. The second is that many deleterious mutations are recessive. While the fixation probabilities of these mutations are not much higher than the fixation probabilities of additive mutations with similar effects, the contribution of recessive mutations to diversity is expected to be higher than that of additive mutations because they fluctuate at higher frequencies and for longer times in the population.

Eq.(32) of the Appendix – L was not previously defined in the Appendix. Assuming that L is the number of sites along the chromosome, the sum must be from 1 to L/2 since the focal site is in the center of the chromosome. Also, in the integral, the term “(r)” must be a subscript.

L327 in Appendix - The "r" is used to represent both the recombination rate and the rate of accumulation of deleterious mutations along the main text and the Appendix, which can lead to confusion.

Reviewer #2: The manuscript by Buffalo & Kern models linked selection across the human genomes using theory from quantitative genetics. Specifically, existing theory (e.g. Stantiago & Caballero 1995; 1998; 2016) describes how selection will influence Ne at a neutral locus linked to a deleterious locus by accounting for the increased variance in reproductive success due to the selective effects. In some parts of the parameter space, this model simplifies to the other background selection theory. However, in the case of weaker selection coefficients, classical background selection theory does not work well because it does not account for the possibility of deleterious mutations drifting upwards in frequency. The authors show how the present theory discussed here provides a more accurate model of linked selection in that part of the parameter space. The authors implement this theory into an inference framework to quantify background selection across the human genome. They obtain estimates of a number of key parameters—such as the deleterious mutation rate, and the distribution of fitness effects (DFE).

Overall, I enjoyed this paper. There are a lot of parts to this manuscript and I think it could be a significant contribution. Key strengths include a more accurate model of background selection via a clever application of previous theory from Santiago and Caballero, a sophisticated inference framework to estimate the parameters from the models, comparing the performance of the new and old models of background selection using simulations, and the analysis of human data.

However, I have a number of questions that I would like to see addressed before the manuscript is considered further for publication.

Major comments:

1) My biggest concern is that some of the estimates of the DFE do not make much sense and are less consistent with previous studies. More specifically, the authors infer that 75% of noncoding phastCons sites have s around 0.01. Other versions of the results in Figure 4 have qualitatively similar patterns for other annotations of noncoding mutations. However, the estimates in Figure 4 show that coding regions have far fewer mutations under strong selection. While I appreciate that the phastCons regions are indeed likely enriched for deleterious mutations, I’m not sure they are more deleterious than amino acid changing mutations. I also appreciated that synonymous mutations are in the CDS category and likely have very small s. Even still, I don’t think this is enough to fully account for the CDS DFE being less deleterious than the noncoding DFE. These results also appear inconsistent with previous work. Indeed, previous work inferred DFEs on human mouse conserved regions that were less deleterious than those for nonsynonymous mutations (Torgerson et al. 2009 PLoS Genetics). More recently, Dukler et al. (2022), taking a different inference approach, also found that CDS mutations experienced a greater reduction in variation than various noncoding annotations. Thus, I’m not sure the DFEs inferred here are plausible. Indeed, the authors seem to imply that their parameter estimates have considerable uncertainty. When they do the local rescaling to correct for interference, the DFE parameters change a lot. I’m not sure what can or should be done about this. I think this issue could stem from some sort of model misspecification (e.g. not modeling structural variants/transposons/STRs that have some fitness effect that gets sopped up in the other annotations; some of the sites classified as “neutral” actually being deleterious, etc). Maybe greater discussion of this topic and less literal interpretation of the DFE parameters would be in order?

As an aside, I do not think that having the “best” estimates of the DFE parameters is a requirement for the present paper to be important. The B’ value map along the genome is an important enough result, in my view.

2) How dependent are these results on the annotations that are used? For example, with the 6% CADD results, selection occurs on these 6% of sites in the genome while the rest of the genome is assumed to be neutrally evolving and only subject to the effects of linked selection. What would happen if deleterious mutations occurred in regions outside the 6% of sites with the most extreme CADD scores? In other words, and more generally, what would happen if some of the sites being treated as neutrally evolving actually experience deleterious mutations? The rationale for this question is that a number of studies have documented turnover of regulatory sequences in the noncoding parts of the genome (e.g. Ponting et al. 2011 Genome Research; Rands et al. 2014 PLoS Genetics; Huber et al. 2020 PLoS Genetics). As such, comparative genomic annotations (like phastCons) may not be able to detect all the sites under selection.

3) As the authors acknowledge, recently Murphy et al. (2022) also presented a model of background selection across the human genome. Their model has some similarities and differences compared to the model used in the present manuscript. While the authors compare the models to see which one has a higher R^2 with the empirical data, I think a more in-depth comparison would strengthen the present manuscript. Specifically, how correlated are the sets of B-values across the genome? Are there specific regions that depart more than others? Examination of such regions might reveal some significant insights.

4) Line 449: The authors apply their approach to infer parameters across several continental ancestry groups. Overall, they find that the inferred DFEs are fairly similar across populations (Figure 4). I agree with the authors that this is an encouraging sign that their inference is not too confounded by demography. I also applaud the authors for looking at how demography influences selection. However, as human populations share a recent common ancestor, I would not expect the DFEs of s to differ too much between populations. Indeed, recent work suggested the DFEs are quite similar (Huang et al. 2021 MBE). What is different between populations is the demographic history. Thus, this means that the patterns of variation may be different across different populations. I believe this would come through in the B’ values that are inferred for each population. How concordant are they across populations? Regions of the genome with different B-values in certain populations could be illustrative of interesting biology (maybe local adaptation, even). I think some additional investigation on this is warranted.

5) The authors would greatly strengthen the impact of their work if they made a nicely curated set of B’ values along the genome easily available to readers. I think this would be an invaluable resource, especially since the original McVicker B-values are still widely-used.

6) Much of the paper focuses on the “sparse track” and the “full track”. These terms should, at least briefly, be explained clearly in the main text before Figure 2 or the first time they’re used. They appear over and over again, so a reviewer (or reader) might get lost.

Minor comments:

1) Line 42: McVicker et al. (2009) actually fit their model to divergence data among primates. They then looked to see how correlated it is with patterns of diversity. Please consider rephrasing.

2) Line 203: The switch to talking about the deleterious substitution rate appears a bit abrupt. It wasn’t immediately clear to me why we’re making the switch to talk about substitutions. Is this to get an extra piece of information from the data (ie substitutions) so that we can better infer parameters (as implied in equations 6 and 7)?

3) Line 237: I believe Figure 11C should be Figure 1C.

4) Line 282-286: I had a hard time following this section here. How does the negative LD lead to the error?

5) Line 356: Is it that the linked selection effect actually matters most at the Mb scale? Or is it that patterns of polymorphism are so noisy at smaller scales, that it’s impossible to detect? I think these are different things, and some clarification here would help.

6) Line 468: I believe you mean Figure 4A instead of Figure 3A.

7) How do the B’ value maps correlate with recombination rates?

Reviewer #3: In this study, Buffalo & Kern investigate how purifying selection shapes the patterns of genetic diversity along the human genome. To do so, they extend an approach initially developed by McVicker et al (PLoS Genet 2009), and later generalized by Murphy et al (eLife 2022) & others, which attempts to fit the megabase-scale variation in pairwise neutral diversity across the genome using a “background selection” model based on the locations of conserved genomic regions. The key difference in this manuscript, compared to previous work, is that they attempt to properly account for the effects of weak selection. This is a longstanding issue in the background selection literature, but it has been neglected in all previous studies that attempt to fit the background selection model to the human genome.

Buffalo & Kern overcome this issue by employing a theoretical approximation developed by Santiago & Caballero (Genetics 2016), which has previously been shown to capture some of the effects of weak selection on pairwise neutral diversity. The authors use this approximation to develop a genome-wide model for the local reduction in pairwise diversity, B’(x), as a function of the recombination map, and the distribution of deleterious fitness effects in different classes of conserved regions. They propose an efficient approach for fitting this model to observed pairwise diversity data from humans, which allows them to obtain predictions for B’(x), as well as estimates for the corresponding DFEs.

Overall, I think this is a nice study that has the potential to significantly advance our understanding of linked selection in the human genome. While the problems created by weak selection have been recognized for some time, our inability to incorporate them into empirical analyses like this one hampered our ability to determine whether simple models of background selection are sufficient to recapitulate the observed data. I have a couple of technical comments, as well as suggestions for improving the clarity and accuracy of the exposition.

Major points:

(1) I think it would be helpful to acknowledge more of the limitations of the Robertson/SC16 approximation when it is introduced in the main text (starting from Eq. 1). For example, previous work has shown that background selection *can’t* be summarized by a rescaled effective population size (see. e.g. Good et al PLoS Genet 2014, Cvijovic et al Genetics 2018). But a naïve reading of the exposition in the Theory section seems to imply that it *can*, and that the main problem is how to pick the right value.

I think the issue stems from the authors’ proof of Eq. 1 in SI Section 1. In a more detailed calculation of the asexual case, Cvijovic et al (Genetics 2018) showed that this genetic drift mapping only holds on timescales longer much longer than the autocorrelation timescale. This can be useful for some observables (e.g. the pairwise neutral heterozygosity) but not for others (e.g. site frequency spectra or substitution rates) that depend on dynamics on shorter timescales. For example, the functional form of the substitution rate in Eq. 7 is not correct, as studies like Neher and Shraiman (Genetics 2012), Good et al (PLoS Genet 2014), and Melissa et al (Genetics 2022) have shown.

I totally buy the argument that there are practical reasons to use the Robertson/SC16 approximation (e.g. computational efficiency), and that the errors it incurs are tolerable for the present application. But being a bit more up front about this in both the main text and supplement would be helpful, so that readers don’t get confused about what is being claimed theoretically.

(2) I was a bit confused about the definition of a “segment”, first mentioned on line 130. How long are these and how is that length scale chosen? Is it related to the “linkage block” concept described in Good et al (PLoS Genet 2014)? How is it related to the concept of a “window” as used for data fitting?

On one hand, it sounds like the segments must be longer than a single site (otherwise Eq. 6 and 7 would return Nf very close to N). But on the other hand, the definition in Eq. 7 makes it seem like there is a single mutation rate and selection coefficient (rather than the full DFE described elsewhere). Later on in the Methods section (Eq. 14), it seems as if different segments are also used to capture different parts of the DFE. How do we reconcile these seemingly conflicting approximations? A bit more discussion of this issue in the text would be helpful.

(3) On a related point, I was confused about the transition from Eq. 2 to Eq. 6 – why does the sum over segments disappear in going from Eq. 2 to Eq. 6? This could be related to the “locally rescaled B’(x)” mentioned later in the paper, which I was also somewhat confused about. Isn’t the locally rescaled version the “correct” version of Eq. 2? It seems like the most straightforward application of the SC16 model would result in an implicit equation for B’(x) of the form

B’(i) = exp[ - sum_j mu_j s_j (1- 4N B’(j) s_j / (exp(4N B’(j)) – 1 )) * Q_ij^2 / 2 ]

with Eq. 6 and the “locally rescaled B’(x)” corresponding to different approximations that drop certain terms in the sum. Is it possible to just solve this equation directly, e.g., by converting to an iterative formula,

B’(i,n+1) = exp[ - sum_j mu_j s_j (1- 4N B’(j,n) s_j / (exp(4N B’(j,n)) – 1 )) * Q_ij^2 / 2 ]

and iterating to convergence starting from B’(i,0) = 1? If so, this might be easier to motivate than the current version. Alternatively, the functional form of Q_ij is such that we expect very distant segments to contribute little to the sum. One could use this intuition to restrict the sum to some maximum distance, so that full quadratic complexity isn’t needed to calculate B’(i). Depending on the length scale chosen for the segments, it may be that Eq. 6 is doing exactly this. But the differences between the initial B’(x) and the “locally rescaled B’(x)” suggests that the default length scale is neglecting some important contributions in the sum.

(4) The authors return to this issue a bit on line 598, but it felt somewhat strange to read it down there (where it is presented almost as an empirical observation) since it seems to follow directly from how their theory is set up. I also don’t understand the argument on line 616 that one should exclude the segment’s own contribution to B(x). Isn’t that exactly what is included in Eqs. 6 and 7?

(5) I was confused by the discussion of “genic” vs “genetic” variance on lines 276-288. Does “genic” refer to the entire segment? Or for the contribution of a single site? Conversely, does “genetic” refer to the entire genome? Or to the entire segment? Some additional clarity here could be helpful, since it relates to the point below:

(6) I could not find the derivation in the SI for the claim on line 286 that “according to theory, the reductions in diversity should be determined by levels of additive genetic fitness variance that include the contribution of LD”. If I understand the SI derivation correctly, it assumes that different segments are in linkage equilibrium with each other (similar to the “linkage block” ansatz from Good et al PLoS Genet 2014), so it actually refers to the “genic” variance in the sense that it neglects LD between segments (i.e. if “genic” is defined at the segment level).

Conversely, the “Fisher’s fundamental theorem” in Eq. 5 applies for the full genetic variance across all segments (see Eq. C.4 in Good & Desai TPB 2013 for a proof). I was therefore confused about the claim on line 282 that “the equilibrium fitness variance modeled by the SC16 theory is predominantly the additive genic fitness variance”. It would help if the authors could clarify which aspects of the SC16 theory they are referring to here.

(e.g. perhaps the authors are referring to the fact that the reason that Eq. 7 is wrong because it is neglecting LD between mutations? In that case I would still argue that it’s not an accurate model of the genic variance either…)

(7) In the main text, the authors talk about inferring a full DFE (e.g. line 290), but in practice they use a discretized version with weight at a fixed # of selection coefficients. I think that’s reasonable, but it would be helpful to mention this approximation in the main text, and how the errors it introduces might influence the biological conclusions. E.g. if the true DFE is “off grid” (or a continuous distribution like a Gamma distribution), what kind of errors does it create if these data are fit to the logarithmically-spaced grid that the authors employ here? I’m particularly curious about the interpretation of the inferred parameters, given the later discussion about whether the inferred mutation rate is too high or not. Could model misspecification in the DFE be contributing to this issue?

(8) I was confused about the use of the term “possible selective interference” in the section title on line 593. I would have thought that this entire paper was looking at the signature of selective interference.. could the authors define what they mean here more explicitly?

(9) I didn’t understand why the molecular clock was necessary on lines 572-576. Doesn’t the theory provide a prediction for the relative divergence along the genome, so you could compare these relative values rather than absolute ones?

(10) If I understand the description on line 636 correctly, the model misspecification issue in the previous section goes away if you use the locally rescaled B’ estimates – is this more evidence that one should be using the iterative re-estimation method from the beginning (as suggested by comment 3 above)? It is worth spending a whole section on the model misspecification issue if it was caused by an approximation that was mainly motivated by computational efficiency?

More broadly, this is a general issue I had with presenting several different approximation schemes in the main text, in that it was sometimes difficult to remember which ones had which implications for the inferred parameters / model checks, and which ones I should believe the most.

(11) Is it possible to do a final forward-time simulation of the authors’ “best” parameter estimates (after all caveats have been considered) and use those to generate a final set of B’ values? That would eliminate any ambiguity in whether the model approximations are accurate.

Minor:

(12) The statement on lines 117-121 feels a little misleading – as described above, one can only neglect the fitness background of the neutral variant on timescales much longer than the autocorrelation time; this is the same timescale on which the fitness backgrounds equalize in the coalescent approach as well (Nicolaisen and Desai, Genetics 2013). Thus, it’s more a benefit of the separation of timescales approximation, rather than the forward-time approach per se.

(13) Line 205. Neher and Shraiman (Genetics 2012), Good et al (PLoS Genet 2014), and Melissa et al (Genetics 2021) have all proposed methods for calculating the rate of Muller’s ratchet in different regimes, including the ones studied here. It would be good to cite these somewhere in this section (perhaps in the discussion of why Eq. 7 breaks down).

(14) I think the reference to Good and Desai (Genetics 2013) on line 170 should be Good et al (PLoS Genet 2014)?

(15) For clarity, it would be great to use a different looking index (e.g. t) for the sum over generations in Eq. 3.

(16) I didn’t understand the rationale behind the “stopping condition” in line 266. Wouldn’t this always be true if we hadn’t simulated enough replicates to begin with? Why do we not simulate more so we can get an absolute estimate of the error? (this seems to be important for the paragraph below this line).

(17) There were various places where I didn’t understand the indices used in the equations in the Methods section. Eq. 13 is an example (the i is summed over in the exponential, while j is not, but both are arguments of the LHS). If the authors could clarify this notation here and elsewhere that would be helpful.

Signed: Benjamin Good (since many of these comments are technical, feel free to contact me directly if I can help to clarify anything)

**Have all data underlying the figures and results presented in the manuscript been provided?**

Reviewer #1: None

Reviewer #2: Yes

Reviewer #3: Yes

PLOS authors have the option to publish the peer review history of their article (what does this mean?). If published, this will include your full peer review and any attached files.

Reviewer #1: **Yes: **Enrique Santiago

Reviewer #2: No

Reviewer #3: No

---

## [Decision Letter · Decision Letter 1]

19 Jan 2024

Dear Dr Buffalo,

We are pleased to inform you that your manuscript entitled "A Quantitative Genetic Model of Background Selection in Humans" has been editorially accepted for publication in PLOS Genetics. Congratulations!

Please note that two reviewers present a few lingering issues. Although I'll spare you another round of submission and review, I encourage you to consider making these minor changes before uploading your final manuscript files.

Yours sincerely,

Bret Payseur

Academic Editor

PLOS Genetics

Justin Fay

Section Editor

PLOS Genetics

Comments from the reviewers (if applicable):

Reviewer's Responses to Questions

**Comments to the Authors:**

Reviewer #1: I appreciate the authors' efforts in revising the manuscript. They have successfully addressed all the concerns I previously raised, and I agree with their responses and the amendments they have implemented in the revised version. The paper now presents a significant advance over prior research in the field.

I overlooked a minor detail in my initial comments. The authors reference two papers by Bulmer (1971) and Keightley and Hill (1988) in line 152, discussing the reduction of genetic variance of quantitative traits under selection. Bulmer's paper offers a theoretical development of an infinitesimal model of unlinked genes, while Keightley and Hill's paper primarily conducts a simulation study of a linked multilocus system. I believe it would be beneficial to also cite Santiago's work (Genet. Res. 1998) here. Santiago's work extends Bulmer's work to linkage and incorporates the two previous works while also considering other models of selection (house of cards, stabilizing). Furthermore, Santiago's model forms the foundation of the theoretical development of Santiago and Caballero (1998), which is pertinent to your study. This suggestion is not crucial to the main topic, so the decision to include the reference is at the authors' discretion.

E. Santiago

Reviewer #2: The revised manuscript is much improved! The authors did a great job addressing my comments, overall. I was especially pleased to see the new Supplementary Figures presenting interesting aspects and patterns of their B-values. I agree with the authors that interpretation and discussion of some of the outlier regions can be left for future work. The paper (and supplement!) is already quite long and full of results.

While I appreciate the authors’ response to my comment about how their estimated DFEs relate to those in previous work, I’m still not convinced that the current DFE estimates are consistent with prior work or make sense biologically. In Figure 4, in looking at the sparse or the full tracks, the CDS mutations have about 75% of their mass in the neutral bin (s=10^-8). In Figure 4B, YRI also have ~75% of CDS mutations in this neutral bin. The DFE from Boyko et al (2008) puts about 30% of nonsynonymous mutations with s<10^-5. Let’s assume all synonymous mutations are neutral. Then the DFE for CDS using the Boyko DFE would have ~50% mutations being neutral (0.3*+0.7*0.3). That is lower than the ~75% the authors report in Figure 4.

Further (not considering prior work), Figure 4B shows that the DFEs for CDS mutations seem to differ dramatically across populations (as also shown in SI Figures 28 and 29). SI Figures 8 and 12 show that the DFE estimates change pretty dramatically with the local rescaling on the inference or with different grids of s. I do not think that these differences represent anything biological, but instead are limitations of the inference.

As I said in my initial review, I do not think that the DFE needs to be perfectly estimated by the author’s approach for the B-value maps to be meaningful. I think this manuscript is a very important contribution to the field and think the authors have done a tremendous amount of work, even if the DFE estimates are not perfect. So, I think my concern with the DFE estimates could easily be addressed with a few sentences of caveats in the Discussion. I do not think that the current revised version of the manuscript includes enough nuance in this area. I’m not sure I agree that “DFE estimates are relatively stable across references populations” as stated on lines 778-780.

A couple of very minor things:

1) Lines 68-70: “…dynamics at one site is not impacted…” should be “are not impacted…”

2) I was a bit confused in lines 341-343 as to why comparisons to pedigree estimates might not be appropriate. Maybe a little more clarification here would help.

3) Line 670: “non-strongly estimates” should probably be “non-strongly deleterious estimates”.

Reviewer #3: I thank the authors for their work in revising the manuscript. The changes have addressed my major concerns, and I think the manuscript is now ready for publication.

**Have all data underlying the figures and results presented in the manuscript been provided?**

Reviewer #1: None

Reviewer #2: Yes

Reviewer #3: Yes

PLOS authors have the option to publish the peer review history of their article (what does this mean?). If published, this will include your full peer review and any attached files.

Reviewer #1: **Yes: **Enrique Santiago

Reviewer #2: No

Reviewer #3: No

**Data Deposition**

http://datadryad.org/submit?journalID=pgenetics&manu=PGENETICS-D-23-01049R1

**Press Queries**

---

## [Editor Report · Acceptance letter]

8 Mar 2024

PGENETICS-D-23-01049R1 

A Quantitative Genetic Model of Background Selection in Humans 

Dear Dr Buffalo, 

We are pleased to inform you that your manuscript entitled "A Quantitative Genetic Model of Background Selection in Humans" has been formally accepted for publication in PLOS Genetics! Your manuscript is now with our production department and you will be notified of the publication date in due course.

With kind regards,

Livia Horvath

PLOS Genetics

On behalf of:
